# Amidst multiple binding orientations on fork DNA, *Saccharolobus* MCM helicase proceeds N-first for unwinding

Himasha M Perera, Michael A Trakselis*

Department of Chemistry and Biochemistry, Baylor University, Waco, United States

**Abstract** DNA replication requires that the duplex genomic DNA strands be separated; a function that is implemented by ring-shaped hexameric helicases in all Domains. Helicases are composed of two domains, an N- terminal DNA binding domain (NTD) and a C- terminal motor domain (CTD). Replication is controlled by loading of helicases at origins of replication, activation to preferentially encircle one strand, and then translocation to begin separation of the two strands. Using a combination of site-specific DNA footprinting, single-turnover unwinding assays, and unique fluorescence translocation monitoring, we have been able to quantify the binding distribution and the translocation orientation of *Saccharolobus* (formally *Sulfolobus*) *solfataricus* MCM on DNA. Our results show that both the DNA substrate and the C-terminal winged-helix (WH) domain influence the orientation but that translocation on DNA proceeds N-first.

DOI: https://doi.org/10.7554/eLife.46096.001

## Introduction

The hexameric MCM complex is conserved throughout archaea and eukaryotic species as the DNA helicase that unwinds the duplex genome providing leading and lagging strand templates for replication. The MCM proteins themselves are bilobal with a N-terminal domain (NTD) that acts to stabilize binding to single-strand DNA (ssDNA), a C-terminal domain (CTD) that contains the conserved AAA$^+$ (ATPases associated with diverse cellular activities) motor domain that provide energy for translocation and DNA unwinding, and a winged-helix (WH) domain for DNA binding (*Trakselis, 2016*). DNA unwinding proceeds by encircling and translocating along the leading strand in the $3'-5'$ direction, while sterically excluding the lagging strand template (*Kelman et al., 1999*; *Chong et al., 2000*; *Bochman and Schwacha, 2009*).

In eukaryotes, six homologous proteins comprise the MCM2-7 heterohexameric complex (*Yuan et al., 2016*). MCM2-7 interacts with Cdc45 and the GINS heterotetramer (Psf1, 2, 3, Sld5) to form the active unwinding CMG complex (*Moyer et al., 2006*). GINS binds primarily to the AAA$^+$ CTD of MCM5 bringing in Cdc45 to interact with and close the interface with MCM2, aligning the motor domains into a proper configuration for activity (*Costa et al., 2011*). Archaea have a single MCM protein that is equally homologous to the six eukaryotic MCM2-7 proteins (*Makarova et al., 2012*; *Goswami et al., 2015*), and in contrast to eukaryotes, the archaeal MCM helicase is active on its own in vitro and does not require accessory proteins for robust DNA unwinding (*Chong et al., 2000*; *Marinsek et al., 2006*). Archaeal MCM forms a homohexameric complex but can also interact with orthologs of Cdc45 (RecJ) and the GINS (GINS23, GINS15) complex to stimulate the helicase activity further (*Yoshimochi et al., 2008*; *Lang and Huang, 2015*; *Xu et al., 2016*), although Cdc45 (*i.e.* GAN) is not required for viability in the euryarchaea,*Thermococcus kodakarensis,* possibly implicating this protein as a redundant nuclease for Okazaki fragment maturation (*Burkhart et al., 2017*).

Loading of the MCM complexes onto and encircling of double-stranded DNA (dsDNA) origins have been a subject of intense experimentation (*Remus et al., 2009*; *Ticau et al., 2015*;

*For correspondence:
michael_trakselis@baylor.edu

**Competing interests:** The authors declare that no competing interests exist.

*Frigola et al., 2017*; *Ticau et al., 2017*), but the consensus origin loaded double hexamer state places the NTDs together with the CTDs facing outwards (*Sun et al., 2013*). This head-to-head orientation achieved during initiation is also conserved with the analogous Large-T antigen of SV40 virus (*Valle et al., 2000*; *Gomez-Lorenzo et al., 2003*). There are still remaining questions as to how the MCM or CMG complex goes from encircling dsDNA to selecting one of the ssDNA strands for translocation. Recent data in the eukaryotic system shows this will involve cyclin dependent kinases (CDK) firing factors, additional components including MCM10, and ATP hydrolysis by MCM subunits to untwist the dsDNA to give two independent CMGs that have encircled opposing strands (*Douglas et al., 2018*). Because the CMG complex translocates 3' to 5', the selection of one strand over the other will dictate whether the two hexamers dissociate from each other or pass over each other for elongation. These two mechanisms will be distinguished by whether the CTD or the NTD, respectively, are leading the way for unwinding. In yeast, the N-first mechanism of CMG translocation has been confirmed, which involves a physical passing of each helicase to regulate origin firing before establishing the replisome progression complex (RPC) (*Georgescu et al., 2017*; *Douglas et al., 2018*), but this has not been confirmed in other species that contain MCM.

The binding orientation of the single archaeal MCM hexamer bound on fork DNA has been shown previously to place the CTD near the fork junction, suggesting a C-first mechanism of unwinding (*McGeoch et al., 2005*; *Rothenberg et al., 2007*; *Costa et al., 2014*). An X-ray structure of an NTD construct of an archaeal MCM shows ssDNA binding orthogonal to the central channel consistent with either N-first or C-first translocation (*Froelich et al., 2014*). C-first translocation for MCM was analogous to the orientation of the homohexameric *Escherichia coli* (*E. coli*) DnaB, which although it has opposite unwinding polarity (5'−3'), also places its CTD RecA motor domain near the duplex region (*Jezewska et al., 1998*). This is now directly challenged by the cryoEM data from higher order eukaryotic systems (*Georgescu et al., 2017*). The strong homology between the archaeal and eukaryotic DNA replication systems would not suggest significant differences in translocation and unwinding mechanisms of the MCM complexes (*Barry and Bell, 2006*).

This report characterizes both the distribution of archaeal MCM binding to the ssDNA regions of fork DNA as well as the translocation orientation of the MCM complex during active unwinding to compare the mechanistic properties between Domains. Many studies have focused on examining the static structural features of helicase binding to DNA or the mechanistic aspects of DNA translocation and unwinding polarity, but few have simultaneously examined both. Using multiple site-specific DNA footprinting techniques, the orientation population distribution of the DNA fork bound *Saccharolobus* (formally *Sulfolobus* [*Sakai and Kurosawa, 2018*]) *solfataricus* (*Sso*) MCM complex was determined. We show that *Sso*MCM can bind both strands of fork DNA in multiple orientations complicating interpretations, however the NTD adjacent to the duplex region (N@duplex) on a 3'-long arm fork is significantly favored, providing more insight into the productive orientation. Binding to fork DNA is affected by the WH domain at the C-terminus that influences the binding orientation. Deletion of the WH domain results in a loss of orientation specificity on 3'-long arm fork substrates mimicking the initial stages of helicase activation. Single-turnover DNA unwinding experiments reveal the stoichiometry of productively bound *Sso*MCM orientations that are influenced by the WH domain and correlate with an N-first translocation and unwinding mechanism. Finally, presteady-state fluorescence resonance energy transfer (FRET) experiments that directly monitor the translocation and unwinding direction of productive *Sso*MCM complexes confirm an N-first mode of unwinding.

## Results

### The orientation distribution of *Sso*MCM is mapped directly on equal arm fork DNA by localized footprinting

Previously, our group and others have shown that *Sso*MCM loads onto fork DNA with the CTD towards the duplex (C@duplex) binding orientation (*McGeoch et al., 2005*; *Rothenberg et al., 2007*; *Costa et al., 2014*), however, its active translocation orientation has yet to be determined. This C@duplex binding orientation has been used to speculate that MCM also translocates in a C-first orientation (*Remus et al., 2009*; *Graham et al., 2011*; *Zhou et al., 2012*; *Bell and Botchan, 2013*; *Costa et al., 2013*; *Costa et al., 2014*; *Miller and Enemark, 2015*; *Martinez et al., 2017*).

However, more recent evidence has shown that when assembled within a leading strand holoenzyme complex, yeast MCM2-7 helicase assembles with the NTD leading the way (N-first) (*Georgescu et al., 2017*). In order to more specifically quantify the binding orientation distribution of *Sso*MCM on model fork substrates, we utilized two separate and specific DNA cleavage strategies.

Single free cysteines within the CTD, at either C642 or C682, were utilized by mutating the other to alanine, releasing an inherent disulfide (*McGeoch et al., 2005*). Either cysteine was then labelled independently with the photoactivatable crosslinker, 4-azidophenacyl bromide (APB). APB attachment at C682 (C642A mutant) provided the greatest signal shift in mobility when crosslinked to DNA (*Figure 1—figure supplement 1*). APB crosslinking to DNA bases is generally non-specific after activation by UV light (*Pendergrast et al., 1992*; *Nodelman et al., 2017*), yet we detected significant crosslinking and subsequent ssDNA cleavage even in the absence of direct UV light (*Figure 1—figure supplement 2*). This was dependent on specific attachment of APB to *Sso*MCM (lanes 5 vs. 3 or 4), and it was enhanced after exposure to UV light and alkaline digestion (lanes 9–11). Overall, *Sso*MCM-APB had many cut sites along the length of both ssDNA substrates favoring positioning at the middle of the ssDNA substrate, implicating nonspecific binding orientation at multiple positions.

To further investigate the orientation of *Sso*MCM on equal arm fork DNA, APB (for crosslinking/digestion) or FeBABE (for a localized hydroxyl radical Fenton footprinting reaction; *Owens et al., 1998*) were conjugated at C682 using *Sso*MCM(C642A) mutant. Cleavage could be induced specifically with UV light/NaOH (APB) (*Figure 1A or D*) or hydrogen peroxide and ascorbic acid (FeBABE) (*Figure 1B or E*) on two separate forks labelled with 5'-Cy3 or 3'-Cy5 at the duplex end. In all situations, multiple cleavage sites were detected on the ssDNA region of the labelled strand (indicated by arrows), suggesting different orientation populations and positioning of *Sso*MCM. *Sso*MCM can bind 3' or 5' ssDNA arms with similar affinities to fork DNA, however when noncomplementary 3' and 5' fork arms are available, there is a preference for binding/encircling the 3'-arm (*Rothenberg et al., 2007*). *Sso*MCM has a significantly lower binding efficiency (~4 fold less) for duplex DNA over fork substrates measured at the single molecule level (*Rothenberg et al., 2007*), essentially eliminating the possibility of *Sso*MCM encircling the duplex region and contributing significantly to cutting the ssDNA arms. Furthermore, anisotropy experiments performed with *Sso*MCM and duplex DNA also show a larger dissociation constant ($K_d$) over fork substrates (*Figure 1—figure supplement 3*), suggesting that *Sso*MCM preferentially binds ssDNA arms of the fork DNA. Moreover, stoichiometric (~1:1 MCM6:DNA) concentration ratios were maintained throughout to promote binding to the highest affinity site and limit nonspecific binding to the duplex region. To test this directly, DNaseI footprinting experiments and Electrophoretic Mobility Shift Assays (EMSA) were performed and confirmed complete DNA binding without protection of the duplex region (*Figure 1—figure supplement 4*). Previously, we have shown that the 5'-excluded strand is protected from ssDNA nuclease digestion upon *Sso*MCM binding (*Graham et al., 2011*) and that titration of large amounts of *Sso*MCM on fork substrates does not compete off the external excluded strand to favor two hexamers binding (*Carney and Trakselis, 2016*). Therefore, the predominate bound species is a stoichiometric single *Sso*MCM hexamer encircling one ssDNA arm and interacting with the other on the exterior surface, but other minor populations also exist.

Using either cleavage agent, there is evidence for footprinting of the CTD of *Sso*MCM towards the duplex end (C@duplex) or the free ends (N@duplex) for either labelled substrate. Cleavage can occur on the encircled strand or the excluded strand consistent with the flexibility of the WH domain to interact with either strand at the fork junction. We quantified and compared the relative footprinting of the CTD delineated by the midpoint of the ssDNA region (*Figure 1C and F*). The midpoint of a ssDNA arm was selected for quantification based on a void in cleavage there and the strong preference for binding ssDNA over duplex DNA at stoichiometric concentrations to describe only binary binding orientations. For either agent (APB or FeBABE), there was a significant ~3:1 preference for placing the CTD closer to the duplex region (C@duplex) independent of which strand is labelled.

## The orientation of *Sso*MCM on asymmetric arm fork DNA by localized footprinting has preference for N@duplex

Although footprinting on equal arm fork DNA favors C@duplex, it is probable that some proportion of *Sso*MCM is encircling the 5'-arm, complicating our analysis and interpretation. Therefore,

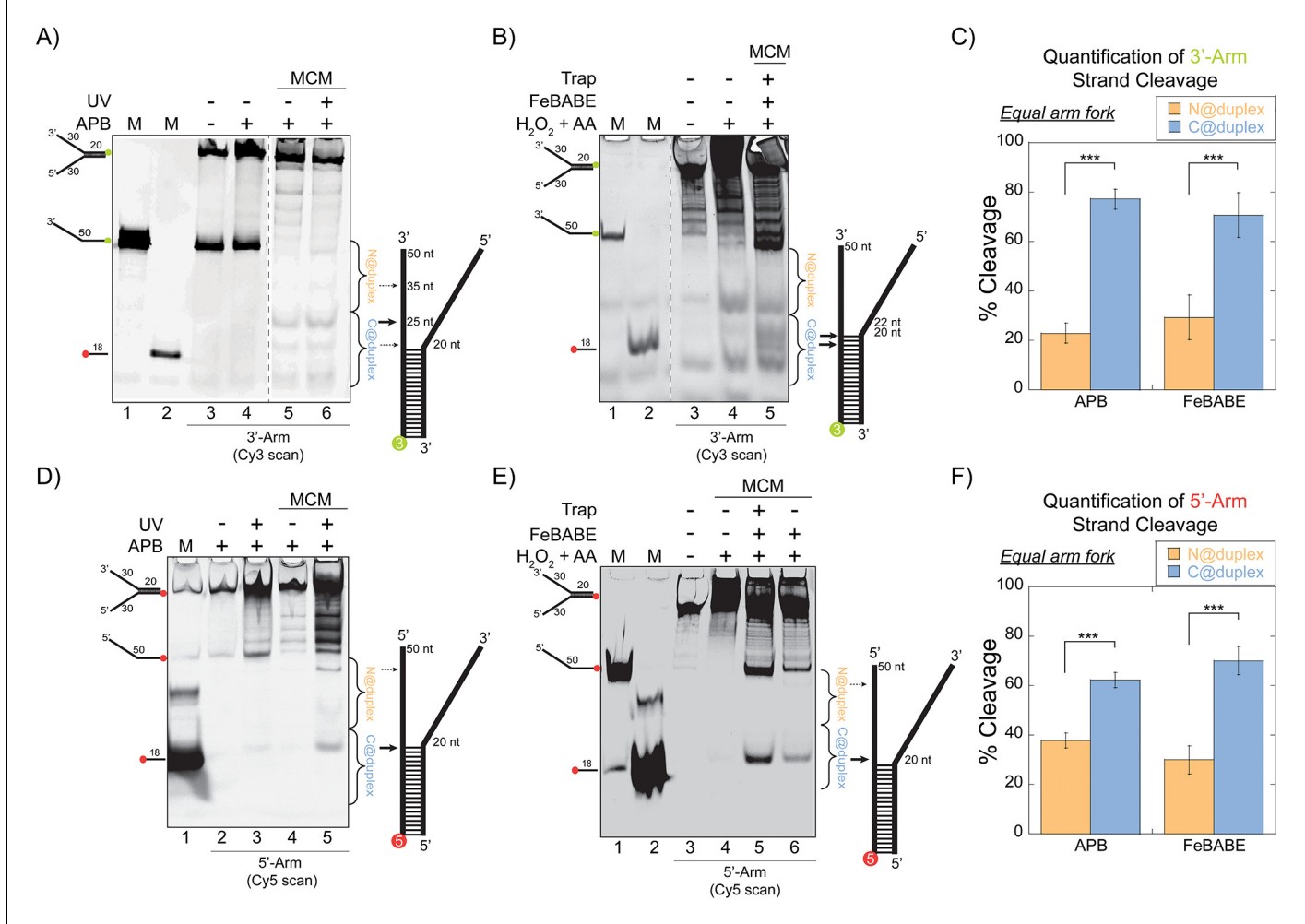

**Figure 1.** SsoMCM orientation mapping onto equal arm fork DNA substrates. (**A**) APB or (**B**) FeBABE orientation mapping of the 3'- encircled strand labelled at the 5'- duplex end with Cy3 on an equal arm fork DNA substrate with a 20 base duplex (DNA164/165-3). SsoMCM was labelled with APB or FeBABE at C682 (within the C-term WH motif) specifically. DNA cleavage was induced and enhanced with UV light [for APB: (**A**), lane 6] or H2O2 and ascorbic acid (**AA**) [for FeBABE; (**B**), lane 5]. Arrow thickness indicates the relative amount and position of DNA cleavage. (**C**) Quantification of the relative amount of DNA cleavage for bases 20-35 or 36-50 from the 5'- end indicate the relative orientation for placing the N-term (N@duplex, orange) or C-term (C@duplex, blue), respectively, closer to the duplex junction for either APB or FeBABE mapping. Similarly, (**D**) APB or (**E**) FeBABE orientation mapping of the 5'- excluded strand labelled at the 3'- duplex end with Cy5 (164-5/165). (**F**) Quantification of the relative amount of DNA cleavage for bases 20-35 or 36-50 from the 5'- end indicate the relative orientation for N@duplex or C@duplex, respectively, closer to the duplex junction for either APB or FeBABE mapping. DNA markers (M) indicate 18 and 50 bases and fork DNA. Error bars represent standard error from 3-5 independent experiments. The products were run on a 20% native PAGE gel. p-values are defined as *< 0.05, **< 0.01 ***<0.001.

DOI: https://doi.org/10.7554/eLife.46096.002

The following figure supplements are available for figure 1:

**Figure supplement 1.** Crosslinking MCM mutants to DNA.
DOI: https://doi.org/10.7554/eLife.46096.003
**Figure supplement 2.** Validation of MCM-APB cleavage on ssDNA.
DOI: https://doi.org/10.7554/eLife.46096.004
**Figure supplement 3.** DNA binding by fluorescence anisotropy.
DOI: https://doi.org/10.7554/eLife.46096.005
**Figure supplement 4.** DNaseI footprinting and EMSA.
DOI: https://doi.org/10.7554/eLife.46096.006

asymmetric arm fork DNA substrates that have a 3'-long arm with different length (0 nucleotide (nt) or eight nt) 5'-arms were designed. Fluorescence anisotropy binding experiments show that *Sso*MCM binds a 5'-long arm substrate with similar affinity to 3'-long arm substrates (*Figure 1—figure supplement 3*). Some archaeal species have a MCM central channel that can occupy both ss and dsDNA (*Fletcher et al., 2003*; *Pape et al., 2003*). Therefore, *Sso*MCM when loaded onto the 3'-long arm fork substrate containing a 0 nt 5'-arm has the possibility of being translocated over the duplex DNA region and then cleaving outside of our boundaries. In order to overcome this, substrates were designed with an 8 nt short 5'-arm. This length was designed to be long enough to prevent translocation over duplex DNA and short enough to prevent helicase loading onto the 5'-arm. It has been previously shown that archaeal MCM requires > 16 nts for productive binding/unwinding (*Haugland et al., 2006*).

Therefore, these orientation mapping experiments were repeated with APB labelled at C682 but limiting the 5'-arm to eight nts to enforce encircling of the 3'-arm. APB footprinting studies of the 3'-long arm substrate instead show that there is nearly a 1.5:1 preference of placing the NTD closer to the duplex region (N@duplex) (*Figure 2A–B*). There is a significant increase and reversed preference for orientating N@duplex for the 3'-long arm fork substrate over the equal arm fork substrate (*Figure 1C*). This suggests that the 5'-long arm either impacts the helicase orientation or that multiple populations of helicases can exist bound on either the 3'- or 5'-arm of the equal arm fork. Therefore, we repeated APB mapping experiments on an opposite 5'-long arm substrate with a shorter 8 nt 3'-arm (*Figure 2C–D*). Here, the footprinting orientations were reversed, with a >3:1 preference for C@duplex (*Figure 2D*). Therefore, on these long arm fork DNA substrates, *Sso*MCM can bind either the 3'- or 5'-arm in both orientations, but the preferred 3'-5' polarity is CTD-NTD.

## The C-terminal WH domain influences the binding orientation of *Sso*MCM on fork DNA

The WH domain at the C-terminus of *Sso*MCM is suggested as a substrate recognition or localization domain (*Aravind et al., 2005*). Moreover, the WH domain in both archaea and eukaryotes is considered important for determining MCM helicase loading and initiation during replication (*Samson et al., 2016a*; *Martinez et al., 2017*; *Goswami et al., 2018*) and mediates DNA binding (*Gaudier et al., 2007*). Thus, we hypothesized that the WH domain may have regulatory effect on directing the orientation of *Sso*MCM helicase on DNA. To determine this, we utilized *Sso*MCM-WH mutant (aa 1–612) with two separate cysteine mutations at the CTD (G452C and S456C) (*Figure 3A*). Footprinting experiments were repeated with APB labelled at either C452 or C456 of *Sso*MCM–WH on equal arm (*Figure 3B*) or 3'-long arm (*Figure 3D*) substrates. The results show a loss of orientation specificity (*Figure 3C and E*) compared with *Figure 1C or 2B*.

As shown above, *Sso*MCM-WH is likely bound on the equal arm fork DNA in at least four populations (two orientations and on either strand). *Sso*MCM WT on equal arm fork substrates (*Figure 1C*) specifically loads C@duplex, but when *Sso*MCM-WH binds the same substrate, it loses a binding preference (*Figure 3C*). When ABP footprinting experiments were repeated with the 3'-long arm substrate, there is a complete loss of orientation specificity on both mutants (*Figure 3E*). These results show that the WH domain of *Sso*MCM influences the binding orientation of this helicase on equal arm fork DNA to place C@duplex but that this WH domain is less important for when engaging ssDNA for translocation.

## Single-turnover DNA unwinding experiments determine relative productive occupancy

Previously, multiple reports have shown that the fraction unwound by *Sso*MCM generally hovers between 0.3 and 0.5 depending on the substrate and conditions (*Barry et al., 2007*; *Graham et al., 2011*; *Graham et al., 2018*). The proportion of *Sso*MCM bound in a productive orientation and state can be determined in a single-turnover DNA unwinding experiment. Single-turnover unwinding conditions were initiated by the simultaneous addition of a 20-fold excess of unlabelled ssDNA and ATP to a prebound *Sso*MCM/DNA complex. The proportion of productive translocating *Sso*MCM hexamers will correlate with the total unwound DNA fraction. Different Cy3 or Cy5 labelled DNA substrates comprised of equal 30 nt fork arms or asymmetric 30 and 8 nt arms were used for unwinding experiments with WT *Sso*MCM (*Figure 4A*). The fork DNA substrate has four possible *Sso*MCM binding

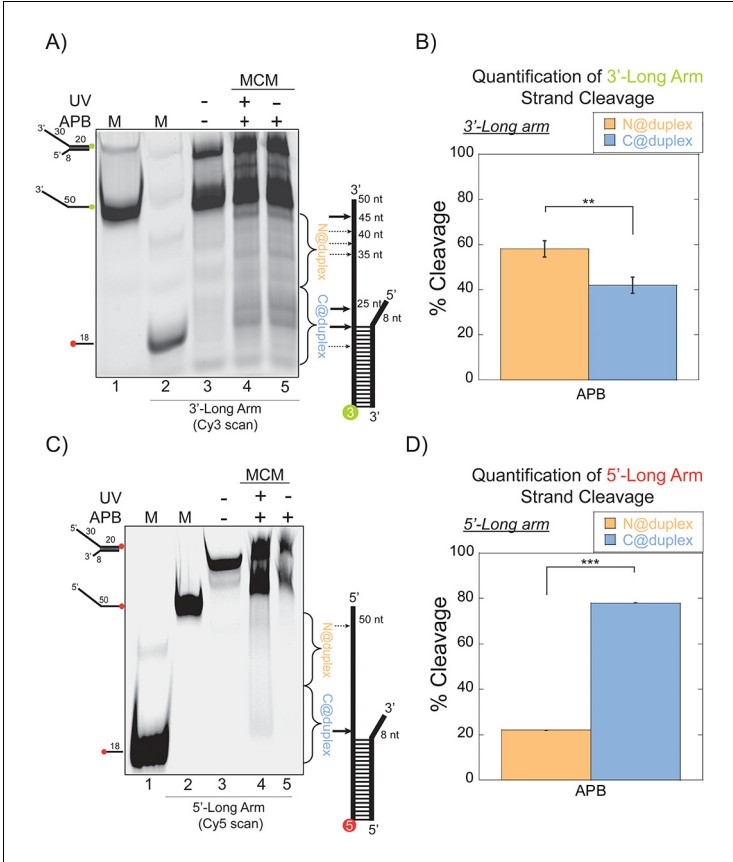

**Figure 2.** *Sso*MCM orientation mapping onto 3'-(DNA171/165-3) or 5'-(DNA172/164-5) long arm fork DNA substrates. (**A**) APB orientation mapping of the 3'-encircled strand labelled at the 5' duplex end with Cy3 on a 3'-long arm fork DNA substrate with a 20 base duplex. *Sso*MCM was labelled with APB at C682 (within the C-term WH motif) specifically. (**B**) Quantification of the relative amount of DNA cleavage for bases 20–35 or 36–50 from the 5'-end indicate the relative orientation for placing the N@duplex (orange) or C@duplex (blue), respectively closer to the duplex junction. Similarly, (**C**) APB orientation mapping of the 5'-excluded strand labelled at the 3'-duplex end with Cy5 on a 5'-long arm fork DNA substrate with a 20 base duplex. (**D**) Quantification of the relative amount of DNA cleavage for bases 20–35 or 36–50 from the 3'-end indicate the relative orientation for N@duplex or C@duplex, respectively, closer to the duplex junction. DNA cleavage was induced and enhanced with UV light [(**A**), lane 4, (**C**), lane 4]. Arrow thickness indicates the relative amount and position of DNA cleavage. DNA markers (**M**) indicate 18 and 50 bases and fork DNA. Error bars represent standard error from 3 to 5 independent experiments. The products were run on a 20% native PAGE gel. p-values are defined as *<0.05, **<0.01 ***<0.001. shorter 8 nt 3'-arm (*Figure 2C–D*). Here, the footprinting orientations were reversed, with a > 3:1 preference for C@duplex (*Figure 2D*). Therefore, on these long arm fork DNA substrates, *Sso*MCM can bind either the 3'- or 5'-arm in both orientations, but the preferred 3'—5' polarity is CTD-NTD.

DOI: https://doi.org/10.7554/eLife.46096.007

orientations (N@duplex or C@duplex on either the 5' or 3'-arms) and unwinds 0.26 ± 0.01 fraction of DNA. Instead, restricting loading to only the 3'-long arm (8 nt 5'-arm) with only two possible orientations significantly increased the unwound fraction to 0.54 ± 0.03. When experiments were repeated with 0 nt at the 5' end, there was 2-fold decrease in unwound product confirming that *Sso*MCM can translocate over the duplex region of the substrates in the absence of any 5'-arm (*Figure 4—figure supplement 1*). Background unwinding on the 5'-long arm (with 0 or 8 nt 3'-arm) displays only 0.08 ± 0.01 or 0.13 ± 0.01 fraction unwound, respectively (*Figure 4—figure supplement 1*). Therefore, an 8 nt 3'-arm is not long enough to facilitate unwinding to any significant degree. Hence, the 3'-long arm (with 8 nt 5'-arm) fork substrate enables the most productive fraction of *Sso*MCM helicases competent for unwinding.

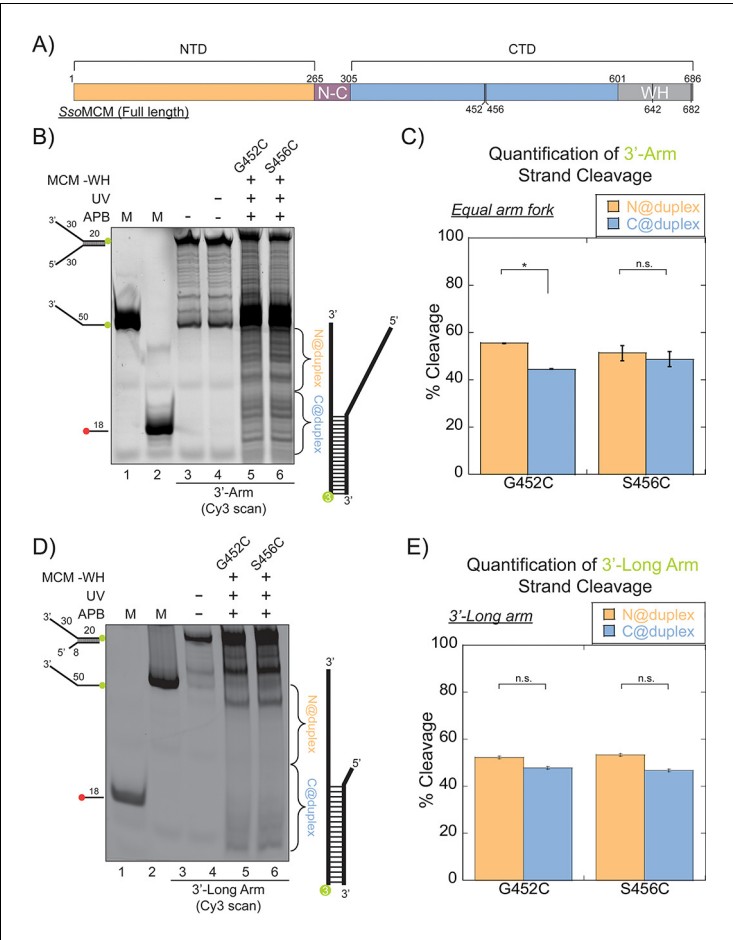

**Figure 3.** SsoMCM (-WH) orientation mapping onto fork DNA substrates. (**A**) Schematic of full length SsoMCM highlighting the NTD (orange), CTD (blue), and the WH domain (grey) and cysteine sites of conjugation used. (**B**) Orientation mapping of SsoMCM labelled at C452 or C456 with APB for the 3'-encircled strand labelled at the 5'-duplex end with Cy3 on a forked DNA substrate with a 20 base duplex. DNA cleavage was enhanced with UV light and the products run on a 20% native PAGE gel. The brackets indicate the regions quantified (bases 20-35 or 36-50) for (**C**) the relative amount and position of DNA cleavage and correspond with the relative orientation of the SsoMCM hexamer; N@duplex (orange) or C@duplex (blue). (**D**) 3'-encircled strand labelled at the 5'-duplex end with Cy3 on a 3'-long arm fork DNA substrate (**E**) the relative amount and position of DNA cleavage and correspond with the relative orientation of the SsoMCM hexamer. DNA markers (M) indicate 18 and 50 bases and fork DNA. Error bars represent standard error from 3-5 independent experiments. p-values are defined as *< 0.05, **< 0.01 ***<0.001. n.s. is not significant.

DOI: https://doi.org/10.7554/eLife.46096.008

As the WH domain was shown above to influence the binding orientation, DNA unwinding was repeated on the fork and 3'-long arm with SsoMCM-WH. Previously, deletion of the WH motif had no effect on DNA binding affinity but significantly increased DNA unwinding in a steady-state experiment (**Barry et al., 2007**). The –WH mutant showed a significant increase in the unwound product with the fork (0.35 ± 0.01) (**Figure 4B**) but a slight decrease with the 3'-long arm (0.46 ± 0.01) (**Figure 4C**) compared with WT. An increased amount of unwound product with the fork substrate suggests a loss in specificity for SsoMCM orientation and correlates with the near equal N@duplex and C@duplex cleavage mapping (**Figure 3C**). The slight decrease in unwound product with the 3'-long arm correlates with the fraction of N@duplex mapped for WT (0.57 ± 0.03) (**Figure 2B**) or –WH (0.52 ± 0.01) (**Figure 3E**) on the same substrate. Therefore, the flexible WH domain influences the population distribution of binding SsoMCM on fork DNA.

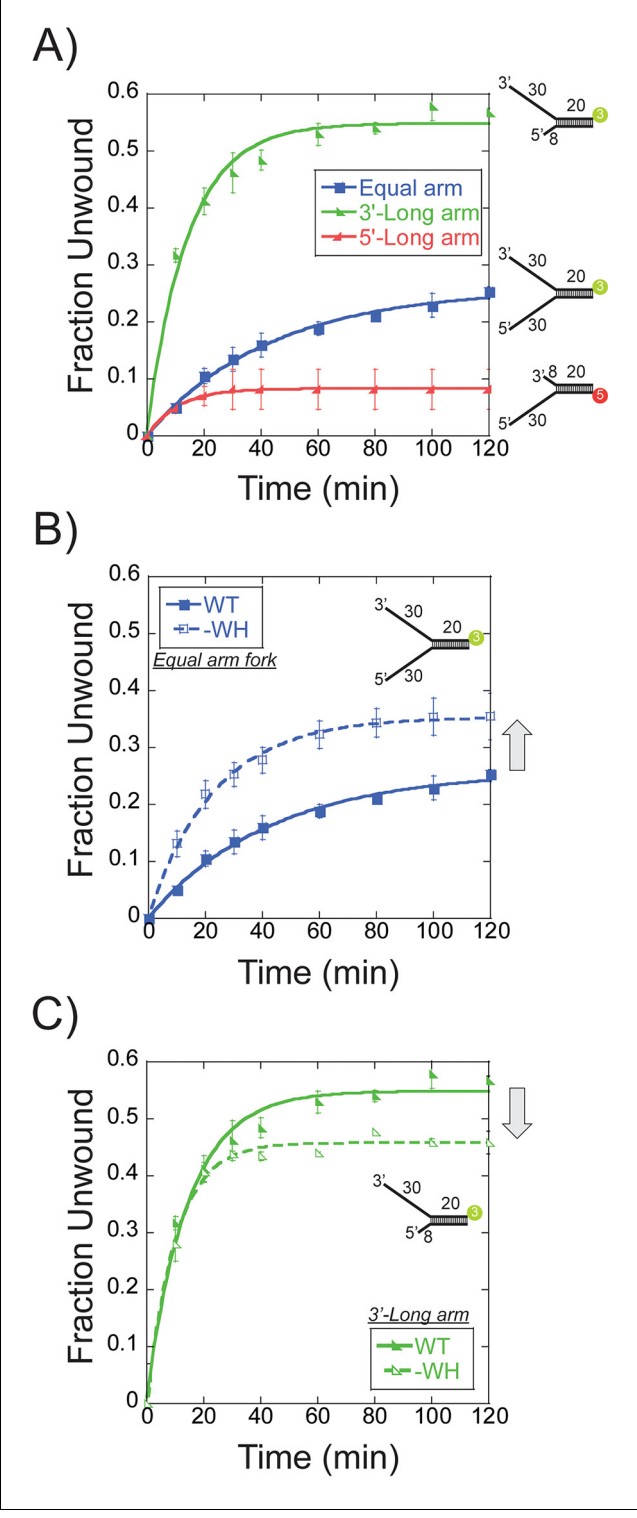

**Figure 4.** Single turnover DNA unwinding. (**A**) DNA unwinding of equal arm (in blue boxes -■-, 5'-Cy3 labelled at the duplex), 3'-long arm (in green lower left triangle -◣-, 5'-Cy3 labelled at the duplex), 5'-long arm (in red lower right triangle -*Sso*MCM WT. Experiments were simultaneously initiated with ATP and an unlabelled ssDNA trap oligo identical to the labelled strand as described in the Materials and methods. (**B**) DNA unwinding of equal arm fork DNA substrate (5'-Cy3 labelled at duplex) with *Sso*MCM WT (in closed boxes -■-) and –WH (in open **boxes** -□-). There is an increase in total DNA unwound with –WH compared to WT (grey arrow). (**C**) DNA unwinding of 3'-long arm fork DNA substrate (5'-Cy3 labelled at duplex) with *Sso*MCM WT (in lower left closed triangle -◣-) and –

*Figure 4 continued on next page*

*Figure 4 continued*

WH (in lower left open triangle -□ -). There is a decrease in the total DNA unwound with –WH compared to WT (grey arrow). Error bars represent standard error from 3 to 5 independent experiments and data was fit to *Equation 3*.

DOI: https://doi.org/10.7554/eLife.46096.009

The following figure supplement is available for figure 4:

**Figure supplement 1.** Single turnover DNA Unwinding.

DOI: https://doi.org/10.7554/eLife.46096.010

Further comparison of DNA unwinding and footprinting results can lead to the identification of the proportion of *Sso*MCM bound in a productive orientation. The fraction unwound for the equal arm fork substrate, 0.26 ± 0.01 (*Figure 4A*), corresponds with a similar footprinting ratio of 0.23 ± 0.03 for N@duplex (*Figure 1C*) implicating an N-first translocation orientation. The fraction unwound for the 3'-long arm fork substrate, 0.54 ± 0.03 (*Figure 4A*), also corresponds with a footprinting ratio of 0.57 ± 0.03 for N@duplex (*Figure 2B*) again correlating with an N-first translocation mechanism.

## Steady-state FRET monitors *Sso*MCM loading on fork DNA at the duplex

To more directly monitor orientation and translocation, we turned to fluorescence assays. Steady-state FRET experiments were designed to qualitatively detect *Sso*MCM binding to forked DNA in a stalled and loaded state from the duplex region. The DNA substrate contains a biotin on the translocating strand (nine bases from the duplex junction) that when bound with streptavidin has been shown to inhibit DNA unwinding (*Graham et al., 2011*) (*Figure 5A*). A fluorescein-dT (FAM) is placed six nts beyond the biotin on the complementary strand and is used to detect FRET upon binding *Sso*MCM labelled at either the N-terminus or C682 with Cy3. *Sso*MCM is able to bind to this substrate in multiple orientations on either the 30mer 3'- or 20mer 5'-strands that will give drastically different FRET signals. The absolute FRET values will depend on the exact spatial location of Cy3 at the N or C-termini and the relative binding orientation distributions. The labelling of *Sso*MCM was controlled by dye stoichiometry and reaction time to give 0.4–0.6 Cy3 labels per *Sso*MCM protein. This puts on average 2–3 Cy3 molecules in the *Sso*MCM hexamer. The experimental FAM quenching result shows an overall small but significant quenching in fluorescence at 518 nm for both labelled *Sso*MCMs (*Figure 5B*) that is consistent with multiple binding populations. The distance of the FAM dye to Cy3 labels near the duplex is modelled to be less than the $R_0$ value for this dye pair (~60 Å) and should be quenched vastly more for one construct over the other if there is a binding preference for either C@duplex or N@duplex. However, results from *Figure 1 and 2* indicate that multiple binding orientations predominate favoring C@duplex when there is a long 5'-arm. Qualitatively, the larger quenching for the C-terminally labelled *Sso*MCM is consistent with a greater distribution that places C@duplex on this semi equal arm fork substrate (*Figure 1*).

## Unwinding directionality determined by presteady-state FRET is confirmed to be N-first

In order to more directly monitor the orientation directionality during unwinding, we changed the experiment setup to monitor presteady-state fluorescence changes in a stopped-flow instrument capable of monitoring loading and translocation of the helicase at 57°C. The 5'-arm was shortened to seven nts to limit loading on that strand and distinguish between translocation orientations solely on the 3'-arm. Binding/loading of *Sso*MCM labelled at C682 or the N-terminus with Cy3 to fork DNA bound by streptavidin showed similar double exponential increases in Cy3 sensitization with rates of 0.57 ± 0.03 $s^{-1}$ and 0.030 ± 0.001 $s^{-1}$ or 0.65 ± 0.02 $s^{-1}$ and 0.043 ± 0.01 $s^{-1}$, respectively. (*Figure 6—figure supplement 1*). Exclusion of ATP in the experiment did not significantly change the exponential results, 0.62 ± 0.01 $s^{-1}$ and 0.043 ± 0.001 $s^{-1}$, for N-terminally labelled *Sso*MCM showing that binding is independent of nucleotide as shown previously (*McGeoch et al., 2005*). Increases in fluorescence are noted for both N and C-terminal labelled *Sso*MCM consistent with both orientations bound.

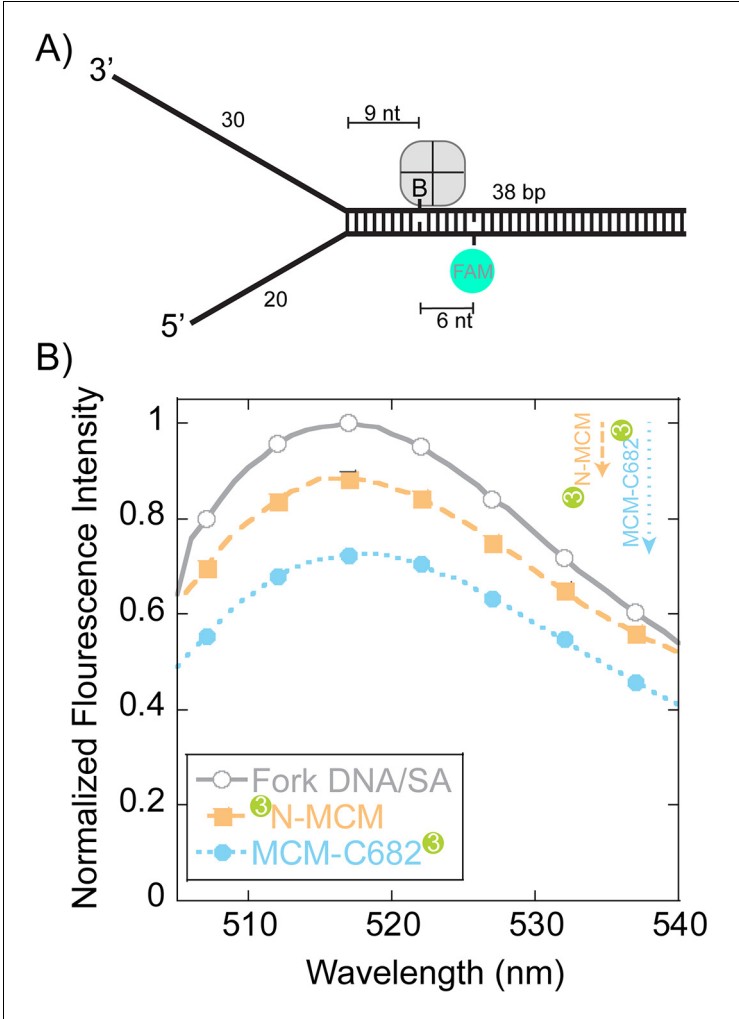

**Figure 5.** Steady-state FRET quenching of *Sso*MCM bound to a fork DNA. (**A**) Schematic of the DNA fork substrate that includes a 30 nt 3'-arm, 20 nt 5'-arm, a biotin placed nine nt from the duplex junction and a dT-FAM placed a further six nt downstream. The $T_m$ of the duplex was calculated to be greater than 75°C. (**B**) Normalized FAM quenching effects upon addition of *Sso*MCM labelled with Cy3 at the N-terminus (closed boxes, orange) or C682 (closed circles, blue). Background changes in fluorescence upon addition of unlabelled *Sso*MCM were subtracted prior to normalization.

DOI: https://doi.org/10.7554/eLife.46096.011

When we preloaded *Sso*MCM on DNA and instead initiated translocation with ATP in the second syringe, we can monitor directionality of movement by FRET up to the streptavidin block. The design of this experiment relies on the longitudinal length of *Sso*MCM (>85 Å), the loading orientation (N@duplex or C@duplex), the placement of dyes at the N or C-terminal ends, and the known 3'−5' unwinding polarity. Although MCM helicases have been shown to displace streptavidin from biotin on the translocating strand, the rate of this displacement in is on the order of hours. Our experimental time courses for these assays are 5 min, where no streptavidin displacement was shown previously (*Graham et al., 2011*). Therefore upon addition of ATP, the MCM helicase will translocate into the duplex region (~9 nts), stall at the streptavidin block, resulting in an increased FRET value only for a fluorescent label on the leading face. The length and sequence of the duplex (36 bases) was designed such that separation of ~9 base pairs would not result in a thermodynamically unstable intermediate at 57°C.

When the Cy3 is labelled at C682, translocation N-first would show a minimal to no increase in fluorescence because of the large distance spanning the length of the *Sso*MCM hexamer; whereas

translocation C-first will show a large increase in FRET upon stalling at the streptavidin block. When the stopped-flow experiment was performed, an initial increase ($0.53 \pm 0.05$ s$^{-1}$) within the first 10 s was noted followed by a slower and more significant decrease in fluorescence ($1.1 \pm 0.2 \times 10^{-3}$ s$^{-1}$) (*Figure 6A*). The first faster increase is consistent with more *Sso*MCM molecules being bound to the DNA template upon addition of ATP (*Figure 6—figure supplement 1*). The second slower change is consistent with dissociation, but not with C-first translocation.

Conversely, when Cy3 is located at the N-terminus, translocation C-first would show little to no change; whereas, translocation N-first would show a large increase in FRET. In the stopped-flow experiment with N-terminal labelled Cy3, there was an initial increase ($0.26 \pm 0.02$ s$^{-1}$) similar to that seen with the label at C682, followed by a larger and slower increase ($1.5 \pm 0.4 \times 10^{-3}$ s$^{-1}$) in fluorescence (*Figure 6B*). The second rate in both experiments (at 57°C) is consistent with the translocation/unwinding rate of *Sso*MCM at 60°C (*Figure 4A*). The single turnover unwinding rate for the 3'-long arm substrate (*Figure 4A*) is $0.07 \pm 0.01$ min$^{-1}$ (or $\sim 1.1 \pm 0.16 \times 10^{-3}$ s$^{-1}$) and is for complete separation of the duplex. The second exponential rate in these presteady-state experiments is

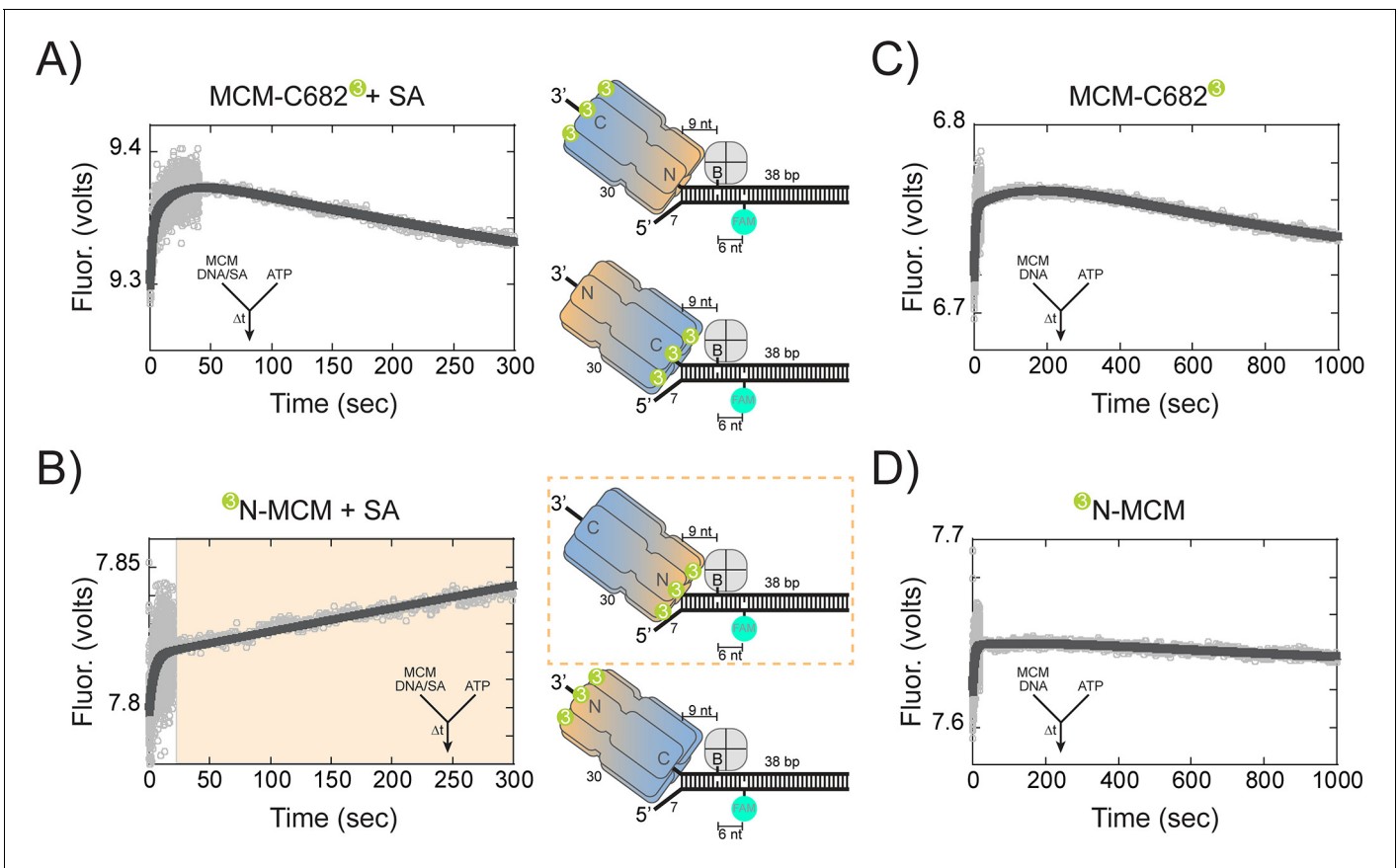

**Figure 6.** Presteady-state stopped-flow measure of SsoMCM translocation/unwinding. SsoMCM (167 nM hexamer) labelled at (**A, C**) C682 or (**B, D**) the N-terminus was preassembled onto 3'-long arm fork DNA (125 nM) with a 30 base 3'-arm and a 7 base 5'-arm in the (**A, B**) presence or (**C, D**) absence of streptavidin (375 nM) as indicated. Translocation was initiated through rapid mixing of ATP (1 mM) and the change in fluorescence above 570 nm was monitored using a split time base at 57 oC. Data was fit to Equation 5. The blue shaded region in (**B**) highlights the N-first translocation mode (boxed). The following figure supplements are available for Figure 6:Figure supplement 1. Presteady-state loading of SsoMCM on DNA. SsoMCM (167 nM hexamer) labelled at (**A**) C682 or (**B**) the N-terminus and preincubated with 1 mM ATP was mixed vs 3'-long arm fork DNA (13 nM) with a 30 base 3-'arm and a 7 base 5'-arm blocked with streptavidin (38 nM) as indicated. (**C**) Experiments in the absence of ATP for SsoMCM labelled at the N-terminus with Cy3. The change in fluorescence above 570 nm was monitored using a split time base at 57 oC. Data was fit to *Equation 5*.
DOI: https://doi.org/10.7554/eLife.46096.012

The following figure supplement is available for figure 6:

**Figure supplement 1.** Presteady-state loading of SsoMCM on DNA.
DOI: https://doi.org/10.7554/eLife.46096.013

extremely similar to the single-turnover experiments and only measures translocation up to nine nts or one fourth of the duplex.

When stopped-flow experiments were performed in the absence of streptavidin, similar initial increases are shown for both C682 ($0.27 \pm 0.01$ s$^{-1}$) and N-terminal ($0.21 \pm 0.01$ s$^{-1}$) labelled *Sso*MCM, but now slower and similar decreases are shown for both labelled constructs ($0.85 \pm 0.05 \times 10^{-3}$ s$^{-1}$ and $1.0 \pm 0.5 \times 10^{-3}$ s$^{-1}$), respectively, consistent with unwinding past the biotin and FAM (*Figure 6C and D*). *Sso*MCM is known to unwind over small adducts such as biotin on the translocating strand (*Graham et al., 2011*) and movement past the FAM label on the excluded strand for both labelled *Sso*MCMs would result in an increase followed by a larger decrease in FRET upon strand separation that would be stochastically blurred in this time scale.

## Translocation orientation on ssDNA determined by presteady-state FRET is consistent with N-first

The fluorescent DNA substrates for the presteady-state FRET experiments were varied to limit duplex length (to 20 bp) and lengthen the single-strand region (to 80 bases) to reduce the possibility of binding and translocating on duplex DNA and complicating our interpretation. Biotin was incorporated four nucleotides prior to the duplex region where a FAM label was placed. Translocation of *Sso*MCM along the ssDNA region would stall when streptavidin was included prior to reaching the duplex but close enough to elicit an increase in FRET when *Sso*MCM is labelled on the leading face with Cy3.

When stopped-flow experiments were repeated with this substrate that included a long 3'-tail, FRET only increased significantly when *Sso*MCM was labelled on the N-terminus with Cy3 and when ATP was included (*Figure 7A*). An initial increase was noted for all experiments consistent with more complex formation as also seen in *Figure 6*. The second rate constant, $5.4 \pm 0.2 \times 10^{-3}$ s$^{-1}$, represents ssDNA translocation by *Sso*MCM. The ssDNA translocation rate is ~5 fold greater than when DNA unwinding is required for a FRET increase (*Figure 6B*). Similar experiments with a long 5'-tail showed minimal changes in FRET (*Figure 7B*). However, when *Sso*MCM is labelled at C682 with Cy3, there is a small decrease in FRET (at a similar ssDNA translocation rate of $2.0 \pm 0.3 \times 10^{-3}$ s$^{-1}$) consistent with the C-terminus moving away from the FAM label in a 3' to 5' manner. No significant change was noted when Cy3 was labelled at the N-terminus on this substrate.

Therefore, our results show that *Sso*MCM can be organized on fork DNA in both orientations with particular probabilities depending on the presence of the excluded strand and the C-terminal WH domain, but translocation and unwinding proceeds N-first in the 3'−5' direction.

## Discussion

*Sso*MCM translocates in the 3'−5' direction, however the orientation during translocation with respect to N or C-first has come under question. Binding assays for archaeal and yeast MCMs on fork or ssDNA show a global orientation preference for the C@duplex (*McGeoch et al., 2005*; *Costa et al., 2014*), however, higher order complexes that include additional yeast replisome components orientate CMG with N@duplex, instead hypothesizing an N-first translocation mechanism (*Georgescu et al., 2017*). This has also been recently confirmed with a x-ray structure of *Sso*MCM bound to ssDNA in an N-first confirmation (*Meagher et al., 2019*). The Costa and Diffley laboratories have provided some guidance that synergizes these two seemingly opposing results as intermediates during the loading, activation, and translocation steps (*Douglas et al., 2018*) that we can better explain mechanistically here.

According to our footprinting assays, there is evidence for placing either CTD or NTD of *Sso*MCM towards the duplex end. Our site specific footprinting experiments show that *Sso*MCM has a 3:1 preference for binding equal arm fork DNA with C@duplex, essentially consistent with our previous results (*McGeoch et al., 2005*; *Rothenberg et al., 2007*). When 3'-long arm fork DNA were used instead, there was a total reversal in orientation preference of 1.5:1 for binding N@duplex. Therefore, we suggest that a large proportion of the C@duplex in the equal arm fork DNA must have been contributed by the *Sso*MCM encircling the 5'- strand DNA (*Figure 1D–F*) but cutting the 3'-strand in proximity. Comparing relative intensities of footprinting for the 5'-strand strand on the equal arm fork substrate (*Figure 1D*) to the 5'-long arm substrate (*Figure 2C*), there is a higher

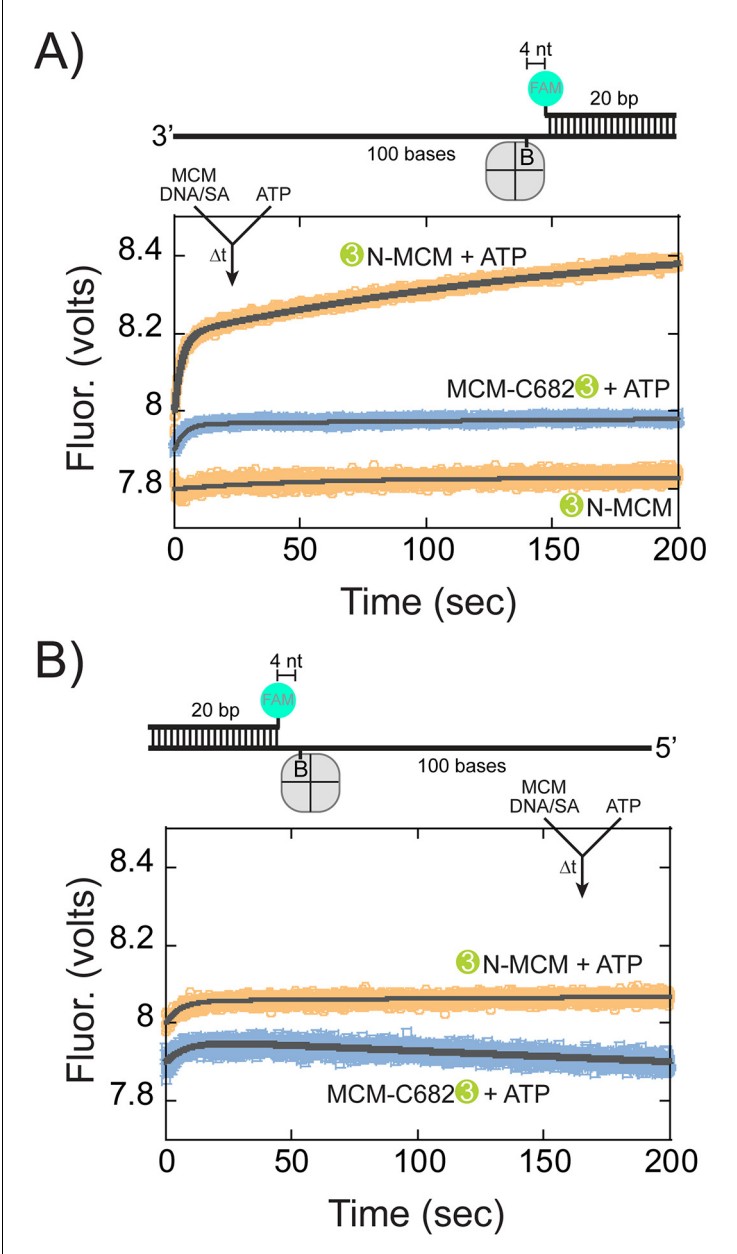

**Figure 7.** Presteady-state stopped-flow measure of *Sso*MCM translocation. *Sso*MCM (120 nM hexamer) labelled at C682 (orange) or the N-terminus (blue) was preassembled onto (**A**) 3'- or (**B**) 5'long arm ssDNA (100 nM) with a flanking 20 bp duplex with 375 nM streptavidin. A FAM label was incorporated at the 5' or 3' end of the duplex, respectively, and four bases from a biotin in the long ssDNA. Translocation was initiated through rapid mixing of ATP (1 mM) and the change in fluorescence above 570 nm was monitored over time at 57°C. Data was fit to *Equation 5*.

DOI: https://doi.org/10.7554/eLife.46096.014

intensity for equal arm fork DNA confirming that *Sso*MCM loaded on the 3'-encircled strand can cut the excluded strand from flexibility rendered by WH domain.

According to previous studies, the C-terminal WH domain of MCM is important for loading the helicase at origins (*Samson and Bell, 2016b*), but the overall DNA binding affinity of *Sso*MCM is not impaired with the WH deletion (*Barry et al., 2007*). Therefore, *Sso*MCM constructs with a deleted WH domain should have no preference for binding DNA in a particular orientation. Footprinting

studies with *Sso*MCM-WH show an almost 1:1 nonselection of loading onto equal arm or asymmetric arm fork substrates in either orientation, suggesting that along with the DNA polarity, the WH domain influences the orientation of *Sso*MCM at the loaded state.

Coupling footprinting fractions with single-turnover unwinding experiments helped determine the orientation fraction of *Sso*MCM involved in active unwinding. The unwinding fraction for equal arm fork DNA substrate is 0.26 and a fractional preference of 0.23 for binding with N@duplex suggesting an N-first translocation orientation. Although *Sso*MCM has a preference for loading on the 3'-arm of a fork substrate (*Rothenberg et al., 2007*), we now show a significant population bound to the 5'-arm, however, *Sso*MCM bound to the 5'-arm is not productive with these substrates. Therefore, a 3'-long arm fork DNA substrate was used to restrict binding/loading onto only the translocating strand. On this substrate, *Sso*MCM has a 0.57 fractional preference for binding with N@duplex and also corresponds with single-turnover unwinding fraction of 0.54 corroborating an N-first unwinding translocation orientation and 3'−5' polarity.

This *Sso*MCM loaded state (C@duplex) is analogous to an initial double hexamer converting to encircling one strand and excluded the other (*Figure 8*). Based on the accepted structure of the MCM double hexamer loaded onto dsDNA origins, the NTDs interact in a head-to-head conformation (*Remus et al., 2009*; *Li et al., 2015*). From that state, there are two possible mechanisms for encircling either the 5'−3' or 3'−5' strands (*Abid Ali et al., 2017*); however in each case, the individual hexamers are still initially orientated in a C@duplex orientation, when both DNA strands are present. Once the excluded strand is melted and displaced outside of the central channel, it can engage with the exterior surface of MCM in a steric exclusion and wrapping (SEW) mechanism (*Graham et al., 2011*). This preloaded and sequestered state is what we have detected in this report

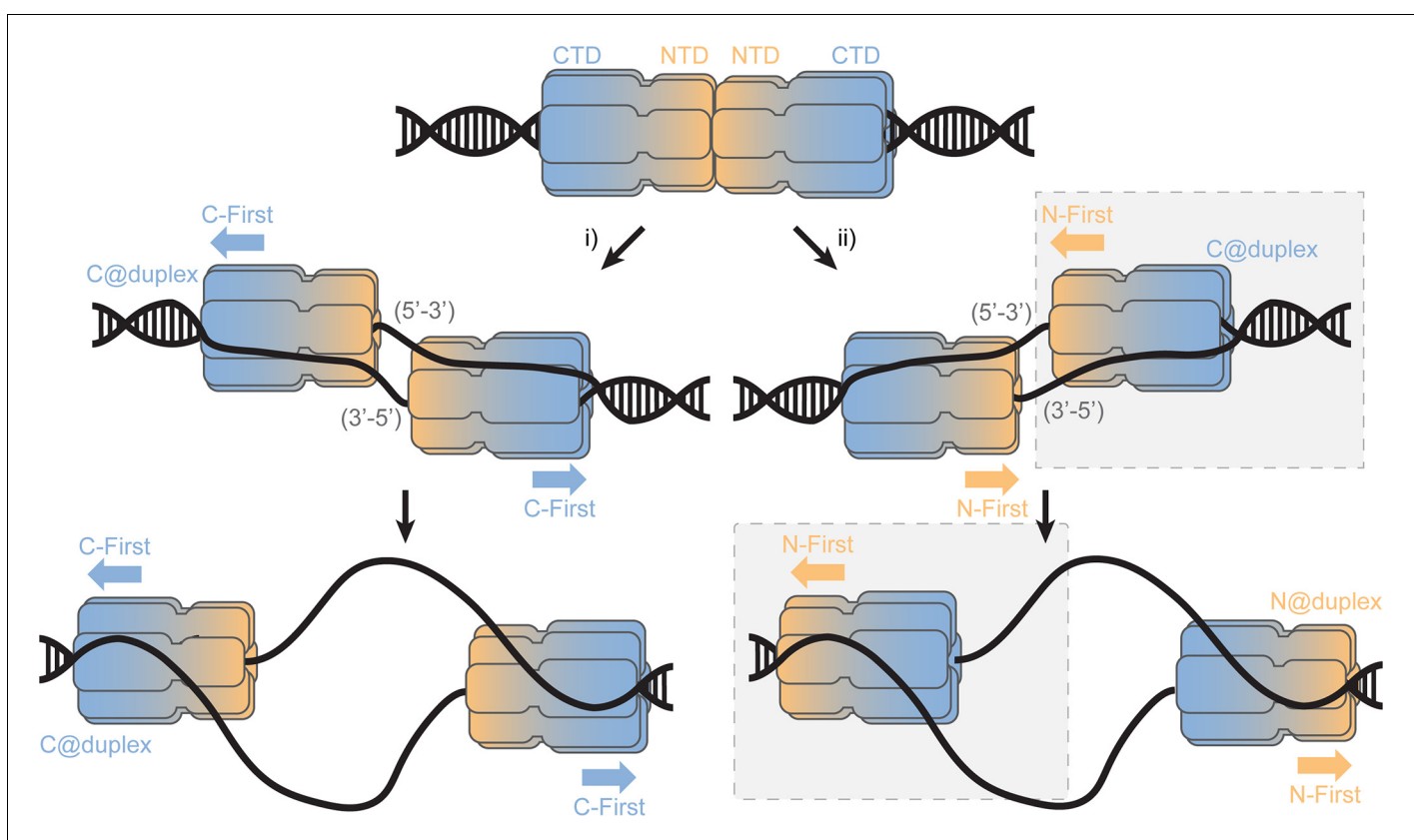

**Figure 8.** SsoMCM loading at origins. Model for loading double hexamer MCM at an origin of replication and the two pathways (i or ii) for encircling the 5'-3' or 3'-5' strands placing the CTD at the duplex (C@duplex). Translocation from (i) would proceed C-first separating hexamers, while translocation from (ii) would proceed N-first bypassing each hexamer. The shaded (grey) box identifies the conformations and states consistent in this report.

DOI: https://doi.org/10.7554/eLife.46096.015

using footprinting studies on equal-arm fork DNA (C@duplex). Interestingly, *Sso*MCM may have a higher affinity for bubble substrates over fork or ssDNA substrates (*Pucci et al., 2004*), which may be achieved through direct double hexamer interactions and/or alternative binding configurations with the bubble region to promote conformational activation.

From there, translocation may proceed in the N-first mode bypassing each hexamer as has recently been observed (*Georgescu et al., 2017*) and indirectly detected (*Douglas et al., 2018*) or in a C-first mode upon separation which had been speculated (*McGeoch et al., 2005*; *Rothenberg et al., 2007*; *Costa et al., 2014*; *Trakselis et al., 2017*). Our presteady-state FRET experiments were performed to directly detect the orientation of the *Sso*MCM hexamer during active translocation and unwinding to be absolutely certain. Using this approach, we could directly monitor the translocation orientation between the NTD of *Sso*MCM and DNA to verify an N-first translocation mechanism.

Combining the results from footprinting, single-turnover unwinding, and presteady-state FRET studies now all support an N-first translocation/unwinding mechanism for *Sso*MCM. After loading at an origin, our results agree with the second pathway (*Figure 8*, ii) for translocation, where two hexamers that have converted to encircling only one DNA strand have to bypass each other to proceed N- first. Similarly, $AAA^+$ papillomavirus E1 helicase which also translocates with 3′−5′ polarity employs a strand exclusion mechanism to unwind DNA proceeding N-first (*Enemark and Joshua-Tor, 2006*; *Lee et al., 2014*). As suggested previously, this would provide an inherent physical control mechanism for DNA unwinding to regulate precise elongation timing (*Li and O'Donnell, 2018*). If pathway i) is incorrectly selected, the N-first 3′−5′ translocation mechanism would inherently block unwinding and render those loaded MCM origins inactive. The consequences of this nonproductive orientation cannot be determined from our current experiments.

The sole selection and encircling of one strand over the other and the conformational steps necessary within the MCM double hexamer remain to be determined and are actively being pursued by a number of laboratories. Some insight into strand selection has be gleamed from a closer examination of the CMG assembly and activation process in eukaryotes (*Douglas et al., 2018*), where ATP binding initiates CMG hexamer separation and early origin melting where DNA becomes underwound in preparation for ssDNA selection. Whether archaeal GINS and Cdc45 influences the binding population orientation on model forks remains to be determined, but the translocation orientation of N-first confirmed here will remain unchanged. Based on a cryo-EM structures of the T7 replisome (*Gao et al., 2019*) and CMG (*Georgescu et al., 2017*) that include ssDNA, it is likely that a helical conformation of DNA will contact multiple subunits in the interior of the hexamer to not only engage one DNA strand to encircle but also for translocation. How the other excluded ssDNA strand slides out between subunits is not yet known but may include contributions of Cdc45 and MCM10 in eukaryotes to remodel CMG and engage that excluded strand on the exterior surface for stability (*Petojevic et al., 2015*; *Mayle et al., 2019*).

## Materials and methods

### Key resources table

| Reagent type (species) or resource | Designation | Source or reference | Identifiers | Additional information |
|---|---|---|---|---|
| Plasmid construct (*E. coli*) | pET30a-SsoMCM (C642A) | *McGeoch et al. (2005)* | | |
| Plasmid construct (*E. coli*) | pET30a-SsoMCM (C682A) | *McGeoch et al. (2005)* | | |
| Plasmid construct (*E. coli*) | pET30a-SsoMCM 1–612 (G452C) | This paper | | Site- directed mutagenesis using primers listed in Materials and methods |
| Plasmid construct (*E. coli*) | pET30a-SsoMCM 1–612 (S456C) | This paper | | Site- directed mutagenesis using primers listed in Materials and methods |
| Expression strain | Rosetta 2 | Novagen | | |

*Continued on next page*

*Continued*

| Reagent type (species) or resource | Designation | Source or reference | Identifiers | Additional information |
|---|---|---|---|---|
| Chemical compound | 4-azidophenacyl bromide (APB) | Sigma-Aldrich | 57018-46-9 | |
| Chemical compound | ATP | Invitrogen | 51963-61-2 | |
| Chemical compound | 1-(p-Bromoacet amidobenzyl) ethylenediamine N, N,N (Fe-BABE) | Dojindo | 186136-50-5 | |
| Chemical compound | DNaseI | New England Biolabs | M0303S | |
| Chemical compound | Streptavidin | Invitrogen | 800-955-6288 | |
| Chemical compound | Cy3 succinimidyl ester | ThermoFisher | 57757-57-0 | |
| Chemical compound | Cy3 maleimide | ThermoFisher | 45-001-273 | |
| Sequence-based reagent | DNA primers and substrates | Sigma-Aldrich and IDT | | Refer to Materials and methods |
| Software, algorithm | Kaleidagraph | www.synergy.com | V4.5 | |

## Materials

ATP was obtained from Invitrogen (Carlsbad, CA). Azidophenacyl bromide (APB) was from Sigma-Aldrich (St. Louis, MO). 1-(p-Bromoacetamidobenzyl) ethylenediamine N, N,N (Fe-BABE) was from Dojindo (Rockville, Maryland). Streptavidin was from Invitrogen (Carlsbad, CA). Cy3 succinimidyl ester and maleimide were from ThermoFisher (Pittsburgh, PA). DNaseI was from NEB (Ipswich, MA). All other materials were from commercial sources and were analytical grade or better. Helicase buffer was used in all unwinding and binding reactions and consists of 125 mM potassium acetate, 25 mM Tris acetate (pH 7.5), and 10 mM magnesium acetate. DNA primers and substrates (*Supplementary file 1*) were all synthesized by Sigma-Aldrich (St. Louis, MO) or IDT (Coralville, IA) and gel purified using crush and soak method (*Maniatis et al., 1989*). Preformed fork substrates: equal arm (DNA164/165), 3'-long arm/5'-(n)nts (n = 0; DNA165/189, n = 8; DNA165/171), 5'-long arm/3'-(n)nts (n = 0; DNA164/190, n = 8; DNA164/172), duplex (DNA180/188), DNA14-B/179 F, DNA14-B/182 F, DNA60-F/202-B and DNA204-F/203-B were heated to 95°C and cooled at a rate of 1 °C /min to room temperature in a PCR instrument.

## Cloning and purification of *Sso*MCM mutants

A cysteine was introduced into *Sso*MCM (1-612) (-WH) at G452C or S456C using a standard Quik-Change protocol (Agilent, Santa Clara, CA) with KAPA HiFi DNA polymerase (KAPA Biosystems, Woburn, MA) with oligos in *Supplementary file 1*. Mutations were initially confirmed by silent mutations to create unique restriction sites and then by the DNA Sequencing Faculty at The University of Texas at Austin (Austin, TX). *Sso*MCM full-length (WT, C642A, and C682A) or 1–612 (-WH: WT, G452C, G456C) were purified as previously described (*McGeoch et al., 2005*; *Graham et al., 2011*). Briefly, autoinduced *Sso*MCM was heat-treated at 70°C for 20 min, and the supernatant was applied to MonoQ, heparin, and S-200 gel filtration columns by use of AKTA Pure (GE Healthscience) to isolate the purified hexameric species.

## Site-specific DNA footprinting using APB

APB was dissolved in 100% DMF at a concentration of 40 mM and then diluted to 4 mM in 20 mM Tris pH 7.5, 75 mM NaCl, 10% glycerol and 20% DMF, in the dark. APB was then added to a sample of *Sso*MCM protein (~10 µM monomer) containing a single cysteine in full length (at either C642 or C682) or in –WH (at either C452 or C456) (in 20 mM Tris [pH 7.5], 75 mM NaCl, 10% glycerol), to

achieve a final concentration of 4 mM APB and 1% DMF. Labelling proceeded for 2–3 hr at room temperature. APB labelled *Sso*MCM was incubated with fluorescent (Cy3 or Cy5 as indicated) fork DNA (150 nM) for 10–20 min in 1x CB buffer (20 mM TrisOAc, 25 mM KOAc, 10 mM MgOAc, 0.1 mg/ ml BSA, 1 mM DTT) in 50 µl volumes (maintaining ~ 1:1 MCM$_6$:DNA ratio). For cross-linking, samples were transferred to silanized cover slips and UV irradiated for 15 s before adding 150 µl of post irradiation buffer (20 mM Tris- HCl [pH 8.0], 0.2% SDS, 50 mM NaCl), vortexed, and placed at 70°C for 20 min. Next, 1 µl of 10 mg/ ml Salmon sperm DNA, 30 µl of 3.0 M NaOAc, 750 µl of ice cold 100% ethanol was added, vortexed, left on ice for 1–2 hr at −80°C. Samples were then spun in microfuge at 4°C, 12,000 rpm for 30 min. The supernatant was discarded, and the pellet was washed twice with ice cold 70% ethanol. Ethanol was removed and the pellets were air dried by inverting on bench for 1 hr and then resuspended in 100 µl: 20 mM NH$_4$OAc, 2% SDS, 0.1 mM EDTA pH 8.0 by vortexing. Samples were spun in microfuge at room temperature for 10 min. Supernatants were transferred into fresh tubes, placed in heat block at 90°C for 2 min. Then, 1 µl of 2 M NaOH was added, vortexed briefly, and incubated at 90°C for 20 min. After incubation, samples were pulse spun, added 101 µl 20 mM Tris- HCl pH 8.0, 1 µl of 2 M HCl, 1 µl of 2 M MgCl$_2$, 480 µl 100% etha- nol, vortexed, and placed at −80°C for 1–2 hr. The samples were pelleted in microfuge at 4°C for 30 min, washed two times with ice cold 70% ethanol, and air dried on bench for 1 hr. The DNA pellet was resuspended with 5 µl of 40% glycerol loading buffer containing Orange G dye for gel loading, run on a 20% TBE- PAGE (native PAGE), and visualized on a Typhoon FLA 9000 imager (GE Healthsciences).

## Site-specific DNA cutting using FeBABE

*Sso*MCM proteins containing a single cysteine at either 642 or 682 for full-length or at 452 or 456 for -WH were dialyzed overnight at 4°C into conjugation buffer (30 mM MOPS, 100 mM NaCl, 1 mM EDTA, 5% glycerol, pH 8.0). Conjugation was performed by mixing 400 µM of FeBABE with 20 µM *Sso*MCM and incubating at 37°C for 1 hr in the dark. After 1 hr incubation, FeBABE-protein conju- gate sample was dialyzed against the cutting buffer (50 mM MOPS, 120 mM NaCl, 0.1 mM EDTA, 10 mM MgCl$_2$, 10% glycerol). Then FeBABE-protein conjugate was mixed with fluorescent DNA (150 nM, as indicated) and incubated at room temperature for 30 min maintaining ~ 1:1 MCM$_6$:DNA ratio. 2.5 µl of ascorbic acid solution (40 mM ascorbic acid, 10 mM EDTA, pH 8.0) was added, vor- texed for 2–3 s, and H$_2$O$_2$ solution (40 mM H$_2$O$_2$, 10 mM EDTA) was added immediately and vor- texed for 2–3 s. The reaction mixture was then incubated for 30 s and quenched by adding Orange G dye loading buffer with 40% glycerol. The samples were electrophoresed on a 20% TBE-PAGE gel and visualized on a Typhoon FLA 9000 imager (GE Healthsciences). Calculation of both the APB and FeBABE footprinting was performed by quantifying the relative density (minus background) for the labelled strand, divided at the midpoint on the ssDNA arm according to the following equation

$$F = \frac{(X@duplex - Control)}{((N@duplex - Control) + (C@duplex - Control))}, X = C, N \qquad (1)$$

A standard two-tailed equal variance student's T-test was used to determine significant differen- ces of C@duplex versus N@duplex. P-values are reported for each experimental condition.

## Single turnover unwinding assays

Single turnover helicase unwinding assays were assembled in helicase buffer with 15 nM concentra- tion of fluorescent forked DNA (as indicated) incubated with 2 µM *Sso*MCM (WT or WH mutant) at 60°C for 5 min before initiating with 2 mM ATP and a 300 nM ssDNA trap (unlabelled strand with the same sequence as the fluorescently labelled strand). Three different fork DNA substrates with a 20 bp duplex region with either Cy3 or Cy5 labels at the duplex end and either 30 nt equal arms or 30 and 8 nt asymmetric arms were used. Unwinding reactions were quenched using an equal volume of quench solution (1.6% SDS, 50% glycerol, 0.1% w/v bromophenol blue, 100 mM EDTA) and an additional 300 nM ssDNA trap at various times. Reactions were placed on ice until loading and were electrophoresed on native 20% TBE-PAGE. The gels were visualized on a Typhoon FLA 9000 imager (GE Healthsciences). The fraction unwound was calculated using the equation:

$$F = \left( \frac{I_{s(t)}}{I_{s(t)} + I_{D(t)}} - \frac{I_{s(0)}}{I_{s(0)} + I_{D(0)}} \right) / \left( \frac{I_{s(b)}}{I_{s(b)} + I_{D(b)}} - \frac{I_{s(0)}}{I_{s(0)} + I_{D(0)}} \right) \qquad (2)$$

where $I_{s(t)}$ and $I_{D(t)}$ are the intensities of the single and double-stranded bands, respectively, at time $t$; subscript $0$ and $b$ indicate equivalent counts at t = 0 and the boiled sample, respectively. The fraction unwound was fit to a single exponential equation as a function of time according to:

$$k = C + Ae^{-kt} \qquad (3)$$

where C is a constant for the amplitude, A is the amplitude change, and k is the rate ($min^{-1}$). The amplitude change denotes the fraction of productive and processive unwinding complexes.

## Fluorescence anisotropy

Anisotropy experiments were performed using a Cary Eclipse Spectrophotometer (Agilent, Santa Clara, CA) in CB buffer. The four forked DNA substrates (with equal arms or asymmetric arms) and the duplex substrate were labelled at the duplex end with either Cy3 at the 5' or Cy5 at the 3' were annealed as described above. Anisotropy measurements were made at each concentration after a 2 min incubation after protein was added. Anisotropy values were collected with a 0.5 s integration time for three consecutive readings. Final values from at least three independent experiments were averaged and fit to a cooperative binding equation:

$$Y = \frac{A_{max} \times [MCM]^n}{((K_d)^n + [MCM]^n)} \qquad (4)$$

in which $Y$ is the measured anisotropy, $A_{max}$ is the maximal anisotropy and $n$ is the Hill coefficient using the Kaleidagraph (Synergy Software, v 4.2).

## DNaseI footprinting

DNaseI footprinting experiments were performed in stoichiometric $MCM_6$:DNA concentration ratios. Equal arm forked DNA substrates (DNA164-5/DNA165) labelled at the duplex end with Cy5 were incubated with $Sso$MCM in 1x CB buffer 15 min at room temperature in 10 µl reaction volumes to facilitate protein-DNA complex formation. The complexes were then digested by 0.1 U/µl DNaseI in 1x DNaseI reaction buffer incubated at 37°C for 30 s. Reaction were then quenched by 5 mM EDTA and heating to 75°C for 10 min. An equal volume of 100% formamide was added and separated on a 20% denaturing PAGE.

## Electrophoretic Mobility Shift Assay (EMSA)

EMSAs were performed in stoichiometric $MCM_6$:DNA concentration ratios. Equal arm forked DNA substrates (DNA164-5/DNA165) labelled at the duplex end with Cy5 were incubated with $Sso$MCM in 1x CB buffer 15 min at room temperature in 10 µl reaction volumes to facilitate protein-DNA complex formation. 2 µl of loading buffer (30% v/v glycerol) was added to the reaction prior to being resolved on 5% native PAGE.

## Presteady-State FRET

Stopped-flow fluorescence experiments were performed on an Applied Photophysics (Leatherhead, UK) SX.20MV in fluorescence mode at a constant temperature of 57°C.

DNA14 was annealed to either DNA179 or DNA182 using to generate two fork substrates with a 30 base 3'-arm and a 20 or 7 base 5'-arm; DNA60 was annealed to DNA202 to give a 3'-long tail substrate; or DNA204 was annealed to DNA203 to give a 5'-long tail substrate. 5'$Sso$MCM(C642A) was labelled at the N-terminus or at C682 with Cy3 as described previously (*McGeoch et al., 2005*). Final concentrations of components after mixing were $Sso$MCM (500 nM or 83 nM hexamer), DNA (50–63 nM), streptavidin (0 or 188 nM), and ATP (0.5 mM), unless indicated otherwise. The samples were excited at 490 nm, and a 570-nm-cutoff filter was used to collect 4000 oversampled data points detecting only Cy3 emission over single or split-time bases. The slits were set at 3 mm for both excitation and emission. At least seven traces were averaged for each experiment and performed

multiple times and on multiple occasions. The observed averaged traces were fit to one, two, or three exponentials using the supplied software. Below is the equation for a double exponential fit:

$$v = a_1 \cdot e^{-k_1 t} + a_2 \cdot e^{-k_2 t} + C \tag{5}$$

where $a$ is the amplitude change, $k$ is the exponential rate, $t$ is time, and $C$ is a constant for the amplitude.

## Acknowledgements

We acknowledge the Baylor Molecular Bioscience Center (MBC) for providing instrumentation and resources aiding this project. We thank Alessandro Costa and Gregory Bowman for helpful discussions.

## Additional information

### Funding

| Funder | Grant reference number | Author |
|---|---|---|
| National Science Foundation | 1613534 | Michael Trakselis |

The funders had no role in study design, data collection and interpretation, or the decision to submit the work for publication.

### Author contributions

Himasha M Perera, Data curation, Formal analysis, Investigation, Visualization, Methodology, Writing—original draft, Writing—review and editing; Michael A Trakselis, Conceptualization, Data curation, Formal analysis, Supervision, Funding acquisition, Investigation, Methodology, Writing—original draft, Project administration, Writing—review and editing

### Author ORCIDs

Himasha M Perera (iD) http://orcid.org/0000-0003-1533-9640
Michael A Trakselis (iD) https://orcid.org/0000-0001-7054-8475

### Decision letter and Author response

Decision letter https://doi.org/10.7554/eLife.46096.019
Author response https://doi.org/10.7554/eLife.46096.020

## Additional files

### Supplementary files

• Supplementary file 1. Table and listing of all DNA sequences and templates used.
DOI: https://doi.org/10.7554/eLife.46096.016

• Transparent reporting form DOI: https://doi.org/10.7554/eLife.46096.017

### Data availability

All data generated or analyzed during the study are included in the manuscript and supporting files.

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
