## [Decision Letter]

**Acceptance summary:**

The polarity by which MCM helicases bind to and translocate along DNA has been highly debated. The current work provides an important biochemical study of an archaeal MCM that reinforces recent findings from homologous systems showing that the N-terminal region of the helicase advances into a DNA fork during unwinding. Collectively, these findings help settle the question of MCM orientation on DNA, providing important insights into replisome organization.

**Decision letter after peer review:**

[Editors’ note: this article was originally rejected after discussions between the reviewers, but the authors were invited to resubmit after an appeal against the decision.]

Thank you for submitting your work entitled "Amidst multiple binding orientations on fork DNA, *Sulfolobus* MCM helicase proceeds N-first for unwinding" for consideration by *eLife*. Your article has been reviewed by a Senior Editor, a Reviewing Editor, and two reviewers. The following individuals involved in review of your submission have agreed to reveal their identity: Eric J Enemark (Reviewer #1).

Our decision has been reached after consultation between the reviewers. Based on these discussions and the individual reviews below, we regret to inform you that your work will not be considered further for publication in *eLife*.

The primary basis for this decision is that neither reviewer felt the data convincingly prove or disprove past work from other laboratories that debate the orientation of MCMs at the replication fork. They note that the overall design of the studies are similar to other experiments that use a double-stranded piece of DNA attached to various single-stranded tails, and which then use the position of the double-stranded region as the means to illustrate MCM orientation. They felt that this approach does not take into account the possibility that the double-stranded region itself helps dictate the MCM:DNA orientation, rather than the polarity of an encircled single strand (they note that the cited Li and O'Donnell structure is also based on this design and therefore is subject to the same concerns). Put another way, the sentiment was that the approach is based on a binary premise, namely, that the ring encircles one single-stranded tail, leaving the N-terminal tier or C-terminal tier will be proximal to the duplex, but that the data actually do not exclude scenarios in which the ring can encircle the duplex region of the DNA as well (which MCM rings are known to bind avidly). This uncertainty dampened confidence in the proposed model.

*Reviewer #1:*

The MCM complex of eukaryotes (MCM2-7) and archaea (MCM) is the "engine" that drives DNA translocation at the replication forks of all eukaryotes and archaea. The manuscript by Perera and Trakselis investigates the orientation of the MCM hexamer of the archaeal organism *Sulfolobus solfataricus (Sso*MCM) in complex with DNA, especially forked DNA. Multiple earlier manuscripts indicated that the MCM complex translocates single-stranded DNA (ssDNA) with the C-terminal domain leading. More recent reports have indicated a reversed orientation such that the MCM complex translocates DNA with the N-terminal domain leading. The translocation orientation is vitally important because it has profound implications for replication initiation where two hexamers of a double-hexamer separate to independent single hexamers to initiate DNA replication in two bidirectional replication forks. The manuscript therefore addresses an important and timely topic, but a few alternative scenarios to the provided interpretation appear highly plausible, and these would continue to confuse assignment of MCM:DNA orientation.

The manuscript provides a likely means to reconcile the opposite translocation orientations observed previously: that the MCM ring is able to bind DNA species in multiple fashions, and that some of these might not be appropriate for DNA translocation. The manuscript develops this conceptual point by evaluating two scenarios of proximity of a duplex DNA portion to the N-terminal side or the C-terminal side of the MCM ring. Based on strand polarities, one of these should be appropriate for 3'-5' DNA translocation (the known polarity of the MCM helicase) while the other would not. The opposite orientation of these binding orientations would thus provide a basis to reconcile previous orientation studies that have led to opposite conclusions on translocation orientation.

However, this treatment does not consider all unproductive species that may have historically confused orientation experiments. In particular, a species where the ring encircles the double-stranded DNA (dsDNA) should be considered. This is not anticipated to be a non-productive unwinding complex due to strand polarity but rather because the MCM ring has not adopted the appropriate strand exclusion topology needed for unwinding (See Fu et al., 2011 for demonstration of strand-exclusion during unwinding). Encircling of dsDNA could occur predominantly at the ATPase tier side or at the N-terminal tier side. In general, if the ring exclusively interacts with the duplex region, the overall 2-fold symmetry of duplex DNA predicts that both binding orientations will be degenerate and should therefore be observed in equal proportion. However, tailed or forked substrates (as used in this study) could develop a specific orientation preference if duplex DNA is encircled at one tier while the ssDNA region binds predominantly at the other tier (for example). Both tiers are known to bind to dsDNA and to ssDNA. Along these lines, the relative affinities of the respective tiers for different structural forms of DNA might have species-specific differences and account for previous differences in assigned orientation. Based on this possibility, the midpoint of the labelled strand may not accurately define the orientation of binding.

To further confuse orientation interpretation, the N-terminal tier of archaeal MCM rings binds single-stranded DNA roughly in the plane of the ring and approximately perpendicular to the channel (Froelich et al., (2014)). Although this binding occurs with a specific polarity, the coplanar MCM:DNA arrangement could be achieved with either 5'- or 3'- extensions to duplex DNA.

Additional comments:

1) Papillomavirus E1 (a close relative of SV40 T-antigen) also forms a dumbbell-shaped double-hexamer with N-terminal sides facing each other. Ultimately, the two hexamers pass over one another in generating two bidirectional replication forks with the N-terminal domains leading in a "strand exclusion" mechanism. The description is essentially identical to "pathway 2" discussed in the present manuscript. Please see Figure S5 of Enemark and Joshua-Tor, (2006) and Figure 7 of Lee et al., 2014. These similarities should be cited. Notably, MCM2-7, MCM, and E1 all belong to the AAA+ family of ATPases and translocate DNA with a 3' to 5' polarity. MCM2-7 and E1 both serve a strand-exclusion helicase function during eukaryotic DNA replication.

2) Discussion section: The manuscript Froelich et al., (2014) reports the structure of the hexameric MCM N-terminal domain bound to single-stranded DNA. As no double-stranded DNA is present, this structure does not illustrate a preferred side for double-stranded DNA. Within its discussion, Froelich et al., (2014) places the structure in a mechanistic framework of DNA translocation with the C-terminal domain leading. The orientation used for this discussion was exclusively based on orientation determinations published in earlier manuscripts (i.e. McGeoch et al., 2005), which showed an orientation with the C-terminal domain leading. This was the best information available at the time. However, the structure itself is consistent with placing duplex DNA at either side, as illustrated in Figure 7B (middle panel) of Froelich et al., (2014), where double-stranded DNA is depicted "above" and "below" the hexameric ring. As such, Froelich et al., (2014) is consistent with both scenarios (placement of the N-terminal domain or the C-terminal domain proximal to double-stranded DNA).

3) For orientation purposes, the manuscript classifies the 3'-extension of a substrate as the "encircled" strand and a 5'-extension as the "excluded" strand (see manuscript's Figure 1). This is the correct assignment when presuming that the helicase preferentially moves towards the double-stranded region due to its 3'-5' polarity. However, the helicase could encircle a 5'-extension strand and follow its intrinsic 3'-5' polarity by moving away from the double-stranded region. The assignment that a given strand is encircled versus excluded implies that the ring somehow recognizes which end possesses a double-stranded region and that the double-stranded region (rather than strand polarities) dictates orientation. The simplest means for this to occur is if the ring actually encircles and binds the dsDNA portion (see main comments). Importantly, this would not be the productive unwinding species because it does not have a "strand excluded" topology.

4) The timescales of unwinding (Figure 4) differ dramatically from those of Figure 7. This could suggest that the generation of a species that is in the appropriate form for unwinding (MCM:DNA in the correct polarity/orientation, strand-excluded, and perhaps other attributes) may generally take a very long time to develop. Such a situation would be consistent with the overall notion that the MCM ring can bind DNA in multiple non-productive configurations. If so, the rapidly formed species examined in Figure 7 would likely represent one of the non-productive conformations or even a mixture of these.

Reviewer #2:

The manuscript aims in addressing the question of the polarity of the MCM helicase from the archaeon *Sulfolobus solfataricus* at the replication fork. The archaeal MCM is composed of two major parts: N-terminal and C-terminal catalytic domains. Using predominantly fluorescence techniques, the study suggests that the helicase N-terminal portion points toward the fork.

The results of this manuscript are in sharp contrast to two previously published studies from the Bell and Ha laboratories (McGeoch et al., 2005; Rothenberg et al., 2007). There is very little discussion of why the studies are reaching completely different conclusions. Based on the data and experiments reported I am not convinced the current manuscript conclusions are correct. Some of my comments are listed below.

The rationale for the study, as outlined in the Introduction, is the eukaryotic CMG EM structure from the Li and O'Donnell laboratories suggesting that the helicase N-terminal domain is pointing toward the fork. However, unlike eukaryotic helicases, most archaeal MCM are active on their own and do not need an assistance of other proteins, such as the CMG complex, for speed and processivity (for example see Schermerhorn et al., 2016). In fact, the archaeal Cdc45 is dispensable for viability (Burkhart et al., 2017). The study does not examine a complex, however, but only the archaeal MCM protein. However, *S. solfataricus* is one of the two organisms studied to date that show an in vitro stimulatory effect on MCM activity by the archaeal GINS and Cdc45 (for example see Xu et al., 2016; Nagata, 2017). Why wasn't the CMG complex included in this study? This is especially important as the rationale for the study and to challenge the previous work was the comparison to the eukaryotic CMG complex. The authors state that "strong homology between the archaeal and eukaryotic DNA replication systems would not suggest significant differences in translocation and unwinding mechanisms of the MCM complexes" in the Introduction. Therefore, the CMG complex should be included in the study.

In subsection “The orientation of *Sso*MCM on asymmetric arm fork DNA by localized footprinting has preference for N@duplex.” it is stated that "it is probable that some proportion of *Sso*MCM is encircling the 5'-arm, complicating our analysis and interpretation. Therefore, these orientation mapping experiments were repeated with APB labelled at C682 but limiting the 5'-arm to 8 bases and forcing encircling of the 3'-arm. It has been previously shown that archaeal MCM requires >15 nucleotides for productive binding/unwinding (Haugland et al., 2006)". There are several issues with this statement. It was shown with many archaeal species, including *S. solfataricus*, that a 5' overhang ssDNA region is not required for helicase activity (for examples see Barry et al., 2007; Chong et al., 2000; Grainge et al., 2003). Why not remove the 5' overhang region altogether? Why leave 8 nucleotides (or 5 nucleotides, in some of the experiments)? If, as stated by the authors, the 5' ssDNA is a problem, it can and should be removed. And, the paper by Haugland et al. describes a unique case among the archaeal species studied. This is the only archaeal MCM that is stimulated by binding to the Cdc6 proteins. The activity of all other archaeal MCMs is inhibited by the Cdc6 proteins. In addition, the study by Haugland et al., did not test a 15 nucleotide ssDNA overhang region as stated in the current manuscript.

Past studies clearly show that the archaeal MCM from different species can displace streptavidin from biotin during translocation along the DNA (for example see: Shin et al., 2013) including a paper from Dr. Trakselis' group on the *S. solfataricus* MCM (Graham et al., 2011). Therefore, it is not clear what the value is of the experiments shown in Figure 5 and Figure 6, as the premise of the experiments is the blocking of DNA unwinding by the biotin/streptavidin complex. If the experimental conditions are different than those previously reported (and therefore no displacement can be observed) then, at a minimum, a control experiment with a biotin trap should be performed to demonstrate this. Dr. Trakselis' group is clearly aware that streptavidin is displaced from biotin by the helicase as they performed such an experiment in a previous publication (Graham et al., 2011).

[Editors’ note: what now follows is the decision letter after the authors submitted for further consideration.]

Thank you for choosing to send your work entitled "Amidst multiple binding orientations on fork DNA, *Sulfolobus* MCM helicase proceeds N-first for unwinding" for consideration at *eLife*. Your letter of appeal has been considered by a Senior Editor and a Reviewing editor, and we are prepared to consider a revised submission without guarantees of acceptance.

The reviewers have evaluated the revised manuscript. Both still share a general concern on the use of double-strand-containing substrates in assigning orientation, and the premise that all species examined are in a "strand excluded" topology. The basis for this concern derives from the fact that MCM rings are capable of interacting with DNA in multiple ways. Biologically, the ring initially encircles double-stranded DNA, and ultimately encircles only one single strand of DNA. Biochemical and structural studies show that the N-terminal tier and the C-terminal AAA+ tier are both able to bind double-stranded DNA and single-stranded DNA, which leads to many possible MCM:DNA structural forms when hybrid single/double-stranded DNA substrates are mixed with an MCM ring. Some alternative binding scenarios with fork DNA include:

1) Species where the ring does not establish a "strand-excluded" topology. Numerous variations are possible-- with the double-stranded DNA portion either at either the N-terminal side, the C-terminal side, or throughout the ring. In the first two cases, it cannot be ruled out that the double-stranded DNA dictates binding orientation rather than encircled strand polarity. The response argues against the presence of these species based on stronger binding to fork substrates versus duplex substrates. This is not an effective argument because the fork substrates could still adopt a topology that is not "strand excluded"-- such as binding of the dsDNA portion at one tier and ssDNA portion(s) at the other(s).

2) Events where the fork binds with an appropriate "strand excluded" topology, but the ssDNA binds exclusively at the N-terminal tier and not the translocating modules (presumably at the AAA+ tier). The N-terminal tier might have a different binding polarity preference than the AAA+ tier, and such a species would then not have the form used for translocation/unwinding.

In the response, the authors state, "…it is our position that duplex DNA would also aid in the positioning of the hexamer along the ssDNA substrate."-- this is the precise concern that the reviewers have when interpreting MCM orientation studies.

For the translocation orientation experiments (Figure 5 and Figure 6), a species in which the ring encircles the 38 bp duplex region at the right cannot be ruled out. Also, species where the ring encircles both arms of the fork at the left cannot be ruled out. Neither is in the correct topology for translocation.

An experiment to determine DNA translocation orientation needs to ensure that the orientation is dictated by the polarity of encircled ssDNA in a binding form derived from the translocating modules (presumably in the AAA+ tier), and that the "lagging strand" (if included) is fully excluded from the ring. These concerns could be resolved by using single-strand-only substrates for the FRET experiment in which the experiment also requires ATPase driven translocation. This could probably be achieved by using a "reasonably long" strand. Such a substrate would remove all the potential points of confusion associated with the possibility of dsDNA driving orientation instead of strand polarity. The attached PDF illustrates the basis for the concern and how this might possibly be resolved with reasonably long single-strand-only substrates. Experiments with this substrate (or an alternative experiment capable of resolving the stated issues) would need to be conducted before a recommendation concerning publication could be made.

---

## [Author Response]

[Editors’ note: the author responses to the first round of peer review follow.]

The primary basis for this decision is that neither reviewer felt the data convincingly prove or disprove past work from other laboratories that debate the orientation of MCMs at the replication fork. They note that the overall design of the studies are similar to other experiments that use a double-stranded piece of DNA attached to various single-stranded tails, and which then use the position of the double-stranded region as the means to illustrate MCM orientation. They felt that this approach does not take into account the possibility that the double-stranded region itself helps dictate the MCM:DNA orientation, rather than the polarity of an encircled single strand (they note that the cited Li and O'Donnell structure is also based on this design and therefore is subject to the same concerns). Put another way, the sentiment was that the approach is based on a binary premise, namely, that the ring encircles one single-stranded tail, leaving the N-terminal tier or C-terminal tier will be proximal to the duplex, but that the data actually do not exclude scenarios in which the ring can encircle the duplex region of the DNA as well (which MCM rings are known to bind avidly). This uncertainty dampened confidence in the proposed model.

A few simple additional experiments and some better explanations in the text can answer all of the reviewers concerns regarding orientation of the *Sso*MCM hexamer during translocation. We want to stress that there are many examples of MCM hexamers binding to duplex DNA and this is an important step during origin binding, but our present studies address the translocation direction during the elongation phase of DNA replication. Even more, we have identified the WH motif at the C-terminal domain of MCM to be important in orientating the bound MCM complex with C@duplex. This explains C@duplex bound orientations (McGeoch, 2005 and Rothenberg, 2008) from our previous work that has confused the field with regards to perceived translocation orientation. In this manuscript, we have directly reconciled binding orientation with that of translocation orientation on fork substrates by quantifying both.

We feel that this updated manuscript has better clarity, explanation, and discussion on the importance of the possible binding orientations that lead to one productive translocation orientation along ssDNA.

Reviewer #1:The MCM complex of eukaryotes (MCM2-7) and archaea (MCM) is the "engine" that drives DNA translocation at the replication forks of all eukaryotes and archaea. The manuscript by Perera and Trakselis investigates the orientation of the MCM hexamer of the archaeal organism Sulfolobus solfataricus (SsoMCM) in complex with DNA, especially forked DNA. Multiple earlier manuscripts indicated that the MCM complex translocates single-stranded DNA (ssDNA) with the C-terminal domain leading. More recent reports have indicated a reversed orientation such that the MCM complex translocates DNA with the N-terminal domain leading. The translocation orientation is vitally important because it has profound implications for replication initiation where two hexamers of a double-hexamer separate to independent single hexamers to initiate DNA replication in two bidirectional replication forks. The manuscript therefore addresses an important and timely topic, but a few alternative scenarios to the provided interpretation appear highly plausible, and these would continue to confuse assignment of MCM:DNA orientation.The manuscript provides a likely means to reconcile the opposite translocation orientations observed previously: that the MCM ring is able to bind DNA species in multiple fashions, and that some of these might not be appropriate for DNA translocation. The manuscript develops this conceptual point by evaluating two scenarios of proximity of a duplex DNA portion to the N-terminal side or the C-terminal side of the MCM ring. Based on strand polarities, one of these should be appropriate for 3'-5' DNA translocation (the known polarity of the MCM helicase) while the other would not. The opposite orientation of these binding orientations would thus provide a basis to reconcile previous orientation studies that have led to opposite conclusions on translocation orientation.However, this treatment does not consider all unproductive species that may have historically confused orientation experiments. In particular, a species where the ring encircles the double-stranded DNA (dsDNA) should be considered.

Previously, we had directly measured at the single molecule level a preference for binding/encircling the 3’-arm over that of the 5’ arm. (Rothenberg et al., 2007). In that same paper, we also measured a significantly lower *Sso*MCM binding efficiency (~4fold less) for duplex DNA over fork substrates. However, this was not articulated well in the previous manuscript to justify the interpretation of our experiments. It is now (Results section).

Furthermore, anisotropy experiments performed with *Sso*MCM and duplex DNA also show a higher dissociation constant (K_d_) over fork substrates (Figure 1—figure supplement 3), suggesting that *Sso*MCM preferentially binds ssDNA arms of the fork DNA. Moreover, titration of large amounts of *Sso*MCM on fork substrates does not compete off the external excluded strand to favor two hexamers binding (Graham et al., 2011. For our footprinting experiments, we are careful to be stoichiometric or sub-stoichiometric (MCM_6_:DNA) to promote binding to the highest affinity site and limit nonspecific binding. Interestingly, we do measure footprinting into the duplex a few bases (Figure 1 and Figure 2), however there is never any indication of cleavage past that towards the duplex end that would be consistent with specific duplex binding. Therefore, the predominate bound species is a stochiometric single *Sso*MCM hexamer encircling the 3’-arm and interacting with the excluded 5’-arm on the exterior surface, but other minor populations also exist.

This is not anticipated to be a non-productive unwinding complex due to strand polarity but rather because the MCM ring has not adopted the appropriate strand exclusion topology needed for unwinding (See Fu et al., 2011 for demonstration of strand-exclusion during unwinding). Encircling of dsDNA could occur predominantly at the ATPase tier side or at the N-terminal tier side. In general, if the ring exclusively interacts with the duplex region, the overall 2-fold symmetry of duplex DNA predicts that both binding orientations will be degenerate and should therefore be observed in equal proportion.

Duplex DNA binding of MCM to origin DNA is known to occur prior to the engagement of a single strand for translocation (as illustrated in Figure 7). However, in this manuscript we are not attempting to answer that question. Rather we are restricting our experiments to test the translocation orientation along ssDNA that is competent for DNA unwinding.

However, tailed or forked substrates (as used in this study) could develop a specific orientation preference if duplex DNA is encircled at one tier while the ssDNA region binds predominantly at the other tier (for example). Both tiers are known to bind to dsDNA and to ssDNA. Along these lines, the relative affinities of the respective tiers for different structural forms of DNA might have species-specific differences and account for previous differences in assigned orientation. Based on this possibility, the midpoint of the labeled strand may not accurately define the orientation of binding.

The midpoint of the ssDNA tail is selected as a convenient place for assigning MCM orientation in a binary fashion. It may not be perfect, but in our footprinting experiments, there is a void in this region that allows us to be confident of this binary distinction in orientation. We were careful to design the length of the ssDNA to be consistent with the site size on DNA and the lateral length of the central channel from available structures. It is possible that part of the N-terminal tier could be encircling duplex DNA (as in Langston and O’Donnell, 2017) however the rest of the complex will be encircling and translocating on ssDNA. This would require further structural/mechanistic confirmation.

To further confuse orientation interpretation, the N-terminal tier of archaeal MCM rings binds single-stranded DNA roughly in the plane of the ring and approximately perpendicular to the channel (Froelich et al., (2014)). Although this binding occurs with a specific polarity, the coplanar MCM:DNA arrangement could be achieved with either 5'- or 3'- extensions to duplex DNA.

With respect to the reviewer, this structure (Froelich et al., 2014), while informative in identifying a MSSB binding motif, does not attempt to address the orientation question. Rather ssDNA binding is visualized with only 7 nucleotides of a 30 base poly dT substrate with a single N-terminal domain protein from another species. We have included this reference in the Introduction. As the reviewer knows, there are several other DNA binding motifs contained within the full length MCM hexamer that would direct ssDNA longitudinally through the entire central channel. Moreover, it is our position that duplex DNA would also aid in the positioning of the hexamer along the ssDNA substrate. A coplanar MCM:DNA arrangement is not expected to occur with full length protein and is instead a better indication of helical engagement of ssDNA within the central channel either for melting of the duplex or translocation (Abid, Diffley and Costa et al., 2017).

Additional comments:1) Papillomavirus E1 (a close relative of SV40 T-antigen) also forms a dumbbell-shaped double-hexamer with N-terminal sides facing each other. Ultimately, the two hexamers pass over one another in generating two bidirectional replication forks with the N-terminal domains leading in a "strand exclusion" mechanism. The description is essentially identical to "pathway 2" discussed in the present manuscript. Please see Figure S5 of Enemark and Joshua-Tor, (2006) and Figure 7 of Lee et al., 2014. These similarities should be cited. Notably, MCM2-7, MCM, and E1 all belong to the AAA+ family of ATPases and translocate DNA with a 3' to 5' polarity. MCM2-7 and E1 both serve a strand-exclusion helicase function during eukaryotic DNA replication.

The reviewer has correctly pointed out the similarities between different organisms that belong to the same AAA+ family of ATPases. We have cited these similarities in the Materials and methods section(Enemark and Joshua-Tor, (2006): Lee et al., (2014)).

2) Discussion section: The manuscript Froelich et al., (2014) reports the structure of the hexameric MCM N-terminal domain bound to single-stranded DNA. As no double-stranded DNA is present, this structure does not illustrate a preferred side for double-stranded DNA. Within its discussion, Froelich et al., (2014) places the structure in a mechanistic framework of DNA translocation with the C-terminal domain leading. The orientation used for this discussion was exclusively based on orientation determinations published in earlier manuscripts (i.e. McGeoch et al., 2005), which showed an orientation with the C-terminal domain leading. This was the best information available at the time. However, the structure itself is consistent with placing duplex DNA at either side, as illustrated in Figure 7B (middle panel) of Froelich et al., (2014), where double-stranded DNA is depicted "above" and "below" the hexameric ring. As such, Froelich et al., (2014) is consistent with both scenarios (placement of the N-terminal domain or the C-terminal domain proximal to double-stranded DNA).

In the previous manuscript, Froelich et al., 2014 was cited under MCM showing a preference for C@duplex (Discussion section). As described by the reviewer #2, this is in accordance with the discussion in Froelich et al., 2014 which speculates that the orientation would be leading the C-terminal domain at the duplex DNA. This sentence is not meant to justify or refute the reviewers own work but to attempt to put the available literature in perspective. We appreciate that their structure does not define the orientation directly and is only built on the available data at the time. We have taken out their reference in that sentence and instead moved to the Introduction to describe this interaction.

3) For orientation purposes, the manuscript classifies the 3'-extension of a substrate as the "encircled" strand and a 5'-extension as the "excluded" strand (see manuscript's Figure 1). This is the correct assignment when presuming that the helicase preferentially moves towards the double-stranded region due to its 3'-5' polarity. However, the helicase could encircle a 5'-extension strand and follow its intrinsic 3'-5' polarity by moving away from the double-stranded region. The assignment that a given strand is encircled versus excluded implies that the ring somehow recognizes which end possesses a double-stranded region and that the double-stranded region (rather than strand polarities) dictates orientation. The simplest means for this to occur is if the ring actually encircles and binds the dsDNA portion (see main comments). Importantly, this would not be the productive unwinding species because it does not have a "strand excluded" topology.

The reviewer is correct in their interpretation here which is why we utilized the terms C@duplex and N@duplex to define orientation which could include encircling either the 3’-arm or the 5’-arm. However, we recognize that some of our labelling in the figures (Encircled or Excluded) could be misinterpreted. Therefore, we have changed these labels to be 3’-arm and 5’-arm for better clarity.

4) The timescales of unwinding (Figure 4) differ dramatically from those of Figure 7. This could suggest that the generation of a species that is in the appropriate form for unwinding (MCM:DNA in the correct polarity/orientation, strand-excluded, and perhaps other attributes) may generally take a very long time to develop. Such a situation would be consistent with the overall notion that the MCM ring can bind DNA in multiple non-productive configurations. If so, the rapidly formed species examined in Figure 7 would likely represent one of the non-productive conformations or even a mixture of these.

The reviewer is actually incorrect here with regards to dramatic differences in rates, but we did not adequately describe that in the original manuscript causing confusion. We better explained these time scales and rates by including s^-1^ value in parentheses. Single turnover DNA unwinding at 60 ^o^C (Figure 4) had a global unwinding rate of 1.1 ms^-1^, while the stopped flow FRET translocation at 57 ^o^C of ~9 base pairs had a rate of 1.5 ms^-1^ (Figure 6). These rates are considered to be equivalent.

Reviewer #2:The manuscript aims in addressing the question of the polarity of the MCM helicase from the archaeon Sulfolobus solfataricus at the replication fork. The archaeal MCM is composed of two major parts: N-terminal and C-terminal catalytic domains. Using predominantly fluorescence techniques, the study suggests that the helicase N-terminal portion points toward the fork.The results of this manuscript are in sharp contrast to two previously published studies from the Bell and Ha laboratories (McGeoch et al., 2005; Rothenberg et al., 2007). There is very little discussion of why the studies are reaching completely different conclusions. Based on the data and experiments reported I am not convinced the current manuscript conclusions are correct. Some of my comments are listed below.

We respectively disagree that there is little discussion throughout. In fact, this whole manuscript attempts to reconcile discrepancies in binding and translocation orientation across multiple papers and groups. McGeoch et al., 2005 describes the major orientation of MCM loaded onto a Y-shaped equal arm DNA substrate (similar to that in Figure 1) placing the C-terminal domain at the double stranded DNA region. We can validate that binding orientation (C@duplex) on fork substrates by site-specific footprinting in Figure 1.

However, McGeoch, (2005) or Rothenberg, (2007) does not monitor the orientation of MCM during translocation (which will indicate the productive orientation) but rather only at a loaded state (which will include both productive and unproductive). The orientation differences between loaded and translocation states are distinguished through this manuscript and are well visualized in Figure 7.

In McGeoch, (2005), since they observe a majority C-terminal domain at the duplex it had been speculated that the translocation would be performed with the motor domain (C-terminal domain) facing the double stranded DNA. However, it is more relevant to monitor the hexamer orientation during translocation. Therefore, we utilized direct (pre-steady state FRET) and indirect (footprinting coupled with single turnover unwinding experiments) methods of determining the translocation orientation of MCM in our current manuscript.

The rationale for the study, as outlined in the Introduction, is the eukaryotic CMG EM structure from the Li and O'Donnell laboratories suggesting that the helicase N-terminal domain is pointing toward the fork. However, unlike eukaryotic helicases, most archaeal MCM are active on their own and do not need an assistance of other proteins, such as the CMG complex, for speed and processivity (for example see Schermerhorn et al., 2016). In fact, the archaeal Cdc45 is dispensable for viability (Burkhart et al., 2017). The study does not examine a complex, however, but only the archaeal MCM protein. However, S. solfataricus is one of the two organisms studied to date that show an in vitro stimulatory effect on MCM activity by the archaeal GINS and Cdc45 (for example see Xu et al., 2016; Nagata, 2017). Why wasn't the CMG complex included in this study? This is especially important as the rationale for the study and to challenge the previous work was the comparison to the eukaryotic CMG complex. The authors state that "strong homology between the archaeal and eukaryotic DNA replication systems would not suggest significant differences in translocation and unwinding mechanisms of the MCM complexes" in the Introduction. Therefore, the CMG complex should be included in the study.

The papers Xu et al., 2016 mostly use *Sulfolobus acidocaldarius* and *Sulfolobus islandicus* in their in vitro and in vivo studies whereas Nagata, 2017 uses *Thermococcus kodakarensis* to show Cdc45 and GINS together as a stable complex that stimulate the MCM helicase activity. None of these used *Sulfolobus (Saccharolobus) solfataricus* proteins. It is very challenging to purify Cdc45 and can only be done from an in vivo endogenous locus in Sac with only nanogram yields from 6L of culture (Xu, 2016). Increased purification yields were found with in vivo expression in *Sulfolobus islandicus*. No purified SsoCdc45 has been shown possible. And we are hesitant to mix species in these experiments as interpretations may be difficult and with artifacts.

We feel that because the *Sso*MCM hexameric complex is active in translocation and unwinding on its own and in the absence of Cdc45 and GINS, that these additional experiments are outside the scope of this manuscript to determine the active translocation orientation.

In subsection “The orientation of SsoMCM on asymmetric arm fork DNA by localized footprinting has preference for N@duplex.” it is stated that "it is probable that some proportion of SsoMCM is encircling the 5'-arm, complicating our analysis and interpretation. Therefore, these orientation mapping experiments were repeated with APB labelled at C682 but limiting the 5'-arm to 8 bases and forcing encircling of the 3'-arm. It has been previously shown that archaeal MCM requires >15 nucleotides for productive binding/unwinding (Haugland et al., 2006)". There are several issues with this statement. It was shown with many archaeal species, including S. solfataricus, that a 5' overhang ssDNA region is not required for helicase activity (for examples see Barry et al., 2007; Chong et al., 2000; Grainge et al., 2003). Why not remove the 5' overhang region altogether? Why leave 8 nucleotides (or 5 nucleotides, in some of the experiments)? If, as stated by the authors, the 5' ssDNA is a problem, it can and should be removed.

The 8-base 5’-tail was included to exclude the possibility raised by reviewer 2 that MCM could be binding to and encircling duplex DNA and cleaving in the duplex region. This 8-base length was designed to be long enough to prevent translocation over duplex DNA and short enough to prevent helicase loading onto the 5’-arm.

However, in response to this reviewer, we performed additional single turnover DNA unwinding (Figure 4—figure supplement 1) and fluorescent DNA binding (Figure 1—figure supplement 3) experiments with 0 nt 5’-arm. These experiments show ~2-fold decrease in unwound product with 0 nt at the 5’-arm compared to a 8 nt 5’-arm. With no 5’ arm, *Sso*MCM can translocate over the duplex region (approx. 40% of the time) which is what we are attempting to prevent. Therefore, asymmetric fork arm substrates with 3’-long arm and a 8 nt short 5’-arm are the best substrate to define the orientation.

And, the paper by Haugland et al. describes a unique case among the archaeal species studied. This is the only archaeal MCM that is stimulated by binding to the Cdc6 proteins. The activity of all other archaeal MCMs is inhibited by the Cdc6 proteins. In addition, the study by Haugland et al., did not test a 15 nucleotide ssDNA overhang region as stated in the current manuscript.

We apologize for this mistake (15 vs 16 nts) in our previous manuscript. The current manuscript has been corrected “It has been previously shown that archaeal MCM requires >16 nucleotides for productive binding/unwinding.” (subsection “The orientation of *Sso*MCM on asymmetric arm fork DNA by localized footprinting has preference for N@duplex”).

Past studies clearly show that the archaeal MCM from different species can displace streptavidin from biotin during translocation along the DNA (for example see: Shin et al., 2013) including a paper from Dr. Trakselis' group on the S. solfataricus MCM (Graham et al., 2011). Therefore, it is not clear what the value is of the experiments shown in Figure 5 and Figure 6, as the premise of the experiments is the blocking of DNA unwinding by the biotin/streptavidin complex. If the experimental conditions are different than those previously reported (and therefore no displacement can be observed) then, at a minimum, a control experiment with a biotin trap should be performed to demonstrate this. Dr. Trakselis' group is clearly aware that streptavidin is displaced from biotin by the helicase as they performed such an experiment in a previous publication (Graham et al., 2011).

Although MCM helicases have been shown to displace streptavidin from biotin on the translocating strand, the rate of this displacement is on the order of hours. Our experimental time courses for these assays are 5 minutes, where no streptavidin displacement was shown previously (Graham et al., 2011). A linear rate of SA displacement was calculated as 0.0067 min^-1^, whereas our measured translocation rate is 0.0015 s-1 or 0.09 min^-1^ (~13-14 fold faster translocation than SA displacement in this manuscript).

Therefore, upon addition of ATP, the MCM helicase will translocate into the duplex region (~9 nts), stall at the streptavidin block, resulting in an increased FRET value only for a fluorescent label on the leading face before displacing SA.

[Editors’ note: the author responses to the re-review follow.]

The reviewers have evaluated the revised manuscript. Both still share a general concern on the use of double-strand-containing substrates in assigning orientation, and the premise that all species examined are in a "strand excluded" topology. The basis for this concern derives from the fact that MCM rings are capable of interacting with DNA in multiple ways. Biologically, the ring initially encircles double-stranded DNA, and ultimately encircles only one single strand of DNA. Biochemical and structural studies show that the N-terminal tier and the C-terminal AAA+ tier are both able to bind double-stranded DNA and single-stranded DNA, which leads to many possible MCM:DNA structural forms when hybrid single/double-stranded DNA substrates are mixed with an MCM ring. Some alternative binding scenarios with fork DNA include:

We appreciate the opportunity to submit a revised manuscript with further supporting data in response to the reviewer comments. We are cognizant to the fact that MCM interacts with DNA in multiple ways; a fact that is recognized by the Title of this manuscript and something we intended to quantify as best we can here. We hope that with the additional requested experiments that were performed, this manuscript is now ready for publication by *eLife*.

1) Species where the ring does not establish a "strand-excluded" topology. Numerous variations are possible-- with the double-stranded DNA portion either at either the N-terminal side, the C-terminal side, or throughout the ring. In the first two cases, it cannot be ruled out that the double-stranded DNA dictates binding orientation rather than encircled strand polarity. The response argues against the presence of these species based on stronger binding to fork substrates versus duplex substrates. This is not an effective argument because the fork substrates could still adopt a topology that is not "strand excluded"-- such as binding of the dsDNA portion at one tier and ssDNA portion(s) at the other(s).

We agree with the reviewers’ concern regarding the ability of the two tiers of *Sso*MCM to accommodate both ss and dsDNA. This is the same concern we set out to address at the beginning of this manuscript by carefully maintaining stoichiometric or sub-stoichiometric MCM_6_:DNA ratios (~1:1) to promote binding of *Sso*MCM to the highest affinity site, which is the ssDNA region (Rothenberg, 2007 and anisotropy data here, Figure 1—figure supplement 3) and limit non-specific binding to duplex regions (Rothenberg 2007) in our site-specific footprinting experiments. This will prevent *Sso*MCM loading onto both ssDNA arms or duplex regions of a given fork substrate at the same time and instead favor encircling of the ssDNA arms adjacent to the duplex.

In order to experimentally support the above hypothesis, we performed new DNaseI footprinting coupled with EMSAs on a fluorescently labeled fork DNA substrate (Figure 1—figure supplement 4). DNaseI can cleave both ss and dsDNA, however the specific activity for cleaving dsDNA is 500-fold more efficient than ssDNA. Protection by *Sso*MCM bound to a dsDNA region will prevent its digestion by DNaseI. Our new data shows that at MCM_6_:DNA @1:1 up to 1:1.5 ratios, the duplex region is not protected in the presence of *Sso*MCM. Therefore, *Sso*MCM is not binding the duplex region specifically. EMSAs confirm that at these ratios, there is complete formation of a MCM_6_:DNA complex.

2) Events where the fork binds with an appropriate "strand excluded" topology, but the ssDNA binds exclusively at the N-terminal tier and not the translocating modules (presumably at the AAA+ tier). The N-terminal tier might have a different binding polarity preference than the AAA+ tier, and such a species would then not have the form used for translocation/unwinding.

We have already addressed this specific situation using our site-specific footprinting experiments coupled with single-turnover unwinding experiments. In our footprinting studies, we have observed two different populations of *Sso*MCM which we term as C@duplex and N@duplex that load onto ssDNA regions (Figure 1 and Figure 2). This nomenclature actually represents at least 4 possible binding confirmations (encircling either ss arm and in both orientations). Using the fraction from the single-turnover unwinding data (Figure 4), we correlated the footprinting data with the productive orientation populations utilized for unwinding to confirm productive unwinding for N@duplex in a 3’-5’ translocation polarity (already described).

In the response, the authors state, "…it is our position that duplex DNA would also aid in the positioning of the hexamer along the ssDNA substrate."-- this is the precise concern that the reviewers have when interpreting MCM orientation studies.

This statement was made in reference specifically for the WH domain influencing the orientation C@duplex or N@duplex (Figure 1 and Figure 2 vs Figure 3) for which we describe.

For the translocation orientation experiments (Figure 5 and Figure 6), a species in which the ring encircles the 38 bp duplex region at the right cannot be ruled out.

Agreed, as this is most likely the confirmation that would be expected upon origin binding (Figure 8, top). The substrate used in Figure 5 and Figure 6 does have 38 bp duplex, however, it also includes a FAM label and Biotin. Once streptavidin is bound to this substrate, the available duplex binding region would shrink substantially.

Nevertheless, we have performed the exact experiments as suggested by the reviewer and incorporated them as Figure 7 in the main text. Of course, we are limited in the duplex length based on the temperature of the activity assays (Tm) but were able to design a 20bp duplex that is stable at the reaction temperature of 57^o^C along with an 80 base ssDNA region that has minimal secondary structure. (See further explanation below).

Also, species where the ring encircles both arms of the fork at the left cannot be ruled out. Neither is in the correct topology for translocation.

This an interesting yet unconventional idea for which the reviewer may be eluding to a modified steric exclusion (MSE, i.e. O’Donnell) or a side channel extrusion (SCE, i.e. Chen and LargeT) modes. Previously, we had used biotin/ streptavidin on either strand of a fork substrate and verified that *Sso*MCM unwinds via a Steric Exclusion mode (Graham, 2011, Figure 1C and D). Even though the *Sso*MCM central channel can accommodate duplex DNA, if *Sso*MCM was to encircle both noncomplementary arms of the fork, unwinding of dsDNA would not be possible via a steric exclusion mode as determined previously with strand specific streptavidin blocks. Whether *Sso*MCM can encircle both noncomplementary strands as one possible mode of binding cannot be ruled out with our current experiments and would be outside the scope of this work.

An experiment to determine DNA translocation orientation needs to ensure that the orientation is dictated by the polarity of encircled ssDNA in a binding form derived from the translocating modules (presumably in the AAA+ tier), and that the "lagging strand" (if included) is fully excluded from the ring.

Again, streptavidin blocks confirmed that the lagging strand is fully excluded from the central channel (i.e. Steric exclusion) during unwinding (Graham et al., 2011).

These concerns could be resolved by using single-strand-only substrates for the FRET experiment in which the experiment also requires ATPase driven translocation. This could probably be achieved by using a "reasonably long" strand. Such a substrate would remove all the potential points of confusion associated with the possibility of dsDNA driving orientation instead of strand polarity. The attached PDF illustrates the basis for the concern and how this might possibly be resolved with reasonably long single-strand-only substrates. Experiments with this substrate (or an alternative experiment capable of resolving the stated issues) would need to be conducted before a recommendation concerning publication could be made.

As suggested by the reviewers and to resolve the concerns described above, new presteady-state FRET experiments were conducted with 80 nucleotide long ssDNA with a biotin (+streptavidin) placed at either end acting as a steric block to translocation polarity (new Figure 8). A FAM label on the adjacent duplex end is utilized for detecting FRET from a translocating *Sso*MCM labelled with Cy3 at the N or C-terminal tiers.

When N-terminally Cy3 labeled *Sso*MCM with 3’-long arm was used, we observed an increase in FRET in the presence of ATP suggesting that *Sso*MCM moves along ssDNA 3’-5’ towards the duplex end, while C-terminally labeled *Sso*MCM does not show an increase in FRET (Figure 8). These results corroborate directly with Figure 7, suggesting a N-first translocation mechanism. Interestingly, a substrate with reverse polarity shows a decrease in FRET only when C-terminally labeled *Sso*MCM is used again suggesting 3’-5’ translocation. These results correlate with a specific translocation orientation polarity preference of the two MCM tiers, where N-tier is facing the 5’-end and C-tier facing the 3’-end. This validates a 3’-5’ translocation polarity for *Sso*MCM, where the N-tier is leading the way.

These additional experiments do not change our initial interpretation that *Sso*MCM has multiple binding orientations but proceeds N-first, but they help to confirm this conclusion.